# Cortical magnification eliminates differences in contrast sensitivity across but not around the visual field

**Michael Jigo[1], Daniel Tavdy[1], Marc M Himmelberg[1,2], Marisa Carrasco[1,2]***

[1]Department of Psychology, New York University, New York, United States; [2]Center for Neural Science, New York University, New York, United States

**Abstract** Human visual performance changes dramatically both across (eccentricity) and around (polar angle) the visual field. Performance is better at the fovea, decreases with eccentricity, and is better along the horizontal than vertical meridian and along the lower than the upper vertical meridian. However, all neurophysiological and virtually all behavioral studies of cortical magnification have investigated eccentricity effects without considering polar angle. Most performance differences due to eccentricity are eliminated when stimulus size is cortically magnified (M-scaled) to equate the size of its cortical representation in primary visual cortex (V1). But does cortical magnification underlie performance differences *around* the visual field? Here, to assess contrast sensitivity, human adult observers performed an orientation discrimination task with constant stimulus size at different locations as well as when stimulus size was M-scaled according to stimulus eccentricity and polar angle location. We found that although M-scaling stimulus size eliminates differences across eccentricity, it does not eliminate differences around the polar angle. This finding indicates that limits in contrast sensitivity across eccentricity and around polar angle of the visual field are mediated by different anatomical and computational constraints.

## Editor's evaluation

This important study presents a thought-provoking challenge to the explanation of sensitivity around the visual field using cortical magnification factors. The evidence supporting this challenge is based on a combination of neuroimaging and psychophysics. The study will be of interest to both basic and medical vision researchers.

***For correspondence:**
marisa.carrasco@nyu.edu

## Introduction

Human visual performance changes throughout the visual field for most visual tasks. Performance is typically best near the fovea and decreases with increasing eccentricity (for reviews, see *Anton-Erxleben and Carrasco, 2013*; *Strasburger et al., 2011*). Contrast sensitivity, a fundamental visual capability, is bandpass near the fovea, peaking at ~4 cycles per degree (cpd) (*Campbell and Robson, 1968*; *Watson and Ahumada, 2005*), and declines with eccentricity (*Hilz and Cavonius, 1974*; *Robson and Graham, 1981*; *Wright and Johnston, 1983*). Contrast sensitivity also varies around polar angle: it is higher along the horizontal than vertical meridian – horizontal-vertical anisotropy (HVA) – and along the lower than upper vertical meridian – vertical meridian asymmetry (VMA) (*Abrams et al., 2012*; *Baldwin et al., 2012*; *Cameron et al., 2002*; *Carrasco et al., 2022*; *Carrasco et al., 2001*; *Hanning et al., 2022a*; *Hanning et al., 2022b*; *Himmelberg et al., 2022b*; *Himmelberg et al., 2020*; *Pointer and Hess, 1989*; *Rosén et al., 2014*). These contrast asymmetries depend upon stimulus eccentricity (*Carrasco et al., 2001*; *Hilz and Cavonius, 1974*; *Himmelberg et al., 2020*; *Robson and*

*Graham, 1981*; *Wright and Johnston, 1983*), spatial frequency (SF) (*Baldwin et al., 2012*; *Cameron et al., 2002*; *Campbell and Robson, 1968*; *Carrasco et al., 2001*; *Himmelberg et al., 2020*; *Rovamo et al., 1992*), and size (*Himmelberg et al., 2020*).

Perceptual polar angle asymmetries can be as pronounced as doubling stimulus eccentricity (*Carrasco et al., 2001*; *Himmelberg et al., 2020*) and are robust across stimulus content modulations. They persist across different stimulus orientations (*Baldwin et al., 2012*; *Carrasco et al., 2001*; *Corbett and Carrasco, 2011*), eccentricities and SFs (*Barbot et al., 2021*; *Cameron et al., 2002*; *Carrasco et al., 2001*; *Himmelberg et al., 2020*; *Rijsdijk et al., 1980*), luminance levels (*Carrasco et al., 2001*), head rotations (*Corbett and Carrasco, 2011*), and sizes (*Himmelberg et al., 2020*), in the presence of distractors (*Carrasco et al., 2001*; *Purokayastha et al., 2021*), across manipulations of covert attention (*Cameron et al., 2002*; *Carrasco et al., 2001*; *Purokayastha et al., 2021*; *Roberts et al., 2018*; *Roberts et al., 2016*; *Talgar and Carrasco, 2002*) and presaccadic attention (*Hanning et al., 2022a*; *Hanning et al., 2022b*), as well as under monocular and binocular viewing conditions (*Barbot et al., 2021*; *Carrasco et al., 2001*).

Importantly, the perceptual asymmetries are retinotopic rather than spatiotopic; when observers rotate their head, the asymmetries shift in line with the retinal location of the stimulus rather than their location in space (*Corbett and Carrasco, 2011*). These asymmetries have been related to optical (*Kupers et al., 2019*), retinal (*Kupers et al., 2022*), and cortical factors (*Benson et al., 2021*; *Himmelberg et al., 2023a*; *Himmelberg et al., 2022b*; *Himmelberg et al., 2021*; *Silva et al., 2018*). For a review, see *Himmelberg et al., 2023b*.

Cortical magnification –the amount of cortical surface area corresponding to one degree of visual angle (mm$^2$/°) – declines with eccentricity (*Benson et al., 2022*; *Engel et al., 1994*; *Himmelberg et al., 2021*; *Horton and Hoyt, 1991*; *Van Essen et al., 1984*) and has been used to link perceptual performance to brain structure (*Duncan and Boynton, 2003*; *Himmelberg et al., 2023a*; *Himmelberg et al., 2022b*; *Rovamo et al., 1978*; *Schwarzkopf et al., 2011*; *Schwarzkopf and Rees, 2013*; *Song et al., 2015*). If performance differences as a function of stimulus location can be attributed to differences in cortical surface area, then performance should be equated when equating stimulus size to the amount of cortical area activated. This can be achieved by enlarging peripheral stimuli (i.e., cortically magnifying, or 'M-scaling') by an inverse proportion to a measure of cortical magnification in the periphery or at different polar angles in the visual field.

Indeed, for many visual tasks (e.g., contrast sensitivity, orientation and SF discrimination, grating acuity, temporal frequency sensitivity, and visual search), magnifying stimulus size such that the spatial extent of the cortical representation is equated eliminates performance differences at different eccentricities (*Figure 1A*; *Anton-Erxleben and Carrasco, 2013*; *Carrasco et al., 1998*; *Carrasco and Frieder, 1997*; *Kitterle, 1986*; *Rovamo and Virsu, 1979*; *Strasburger et al., 2011*). Particularly relevant to this study, grating contrast sensitivity as a function of SF successfully scales with eccentricity (*Hilz and Cavonius, 1974*; *Koenderink et al., 1978*; *Rovamo et al., 1978*; *Virsu and Rovamo, 1979*). These studies support a 'quantitative' hypothesis – the hypothesis that the decline in performance with eccentricity is due to decreasing neural count with eccentricity, given that the density, but not distribution, of V1 neurons is approximately uniform across cortex (and thus visual space) (*Hubel and*

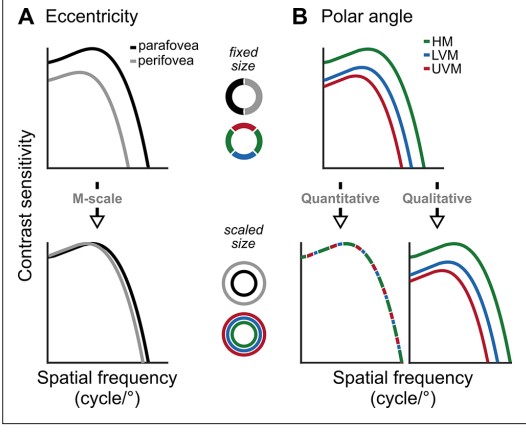

**Figure 1.** Schematic predictions of contrast sensitivity functions (CSFs). (**A**) CSFs decline between the parafovea (2°) and perifovea (6°) for fixed-sized gratings (top row), but differences at low and medium SFs diminish after M-scaling *Virsu and Rovamo, 1979* (bottom row). (**B**) CSFs differ among the horizontal meridian (HM), lower vertical meridian (LVM), and upper vertical meridian (UVM) for fixed-sized stimuli (top row). If polar angle asymmetries derive from differences in neural count among locations, M-scaling will diminish them ('quantitative' hypothesis). Alternatively, if the asymmetries derive from qualitatively different neural image-processing capabilities among locations, then M-scaling will not eliminate them ('qualitative' hypothesis).

*Wiesel, 1977*; *Rockel et al., 1980*). This hypothesis then suggests that visual processing is invariant with location; neurons conduct the same computations, regardless of their receptive field position.

Alternatively, when performance cannot be matched by M-scaling stimulus size (*Figure 1B*), then a 'qualitative hypothesis' is supported, stating that performance differences are mediated both by the cortical representation – and thus neural count – but also by different computations within visual neurons that encode different visual field locations. Indeed, performance does not successfully scale for several visual tasks measuring higher-order dimensions (e.g., numerosity judgments, symmetry detection, and positional relation of image components; for review, see *Strasburger et al., 2011*).

So far, these hypotheses have been supported by different visual tasks. Here, we ask whether they are exclusive for a given task or whether some regions of the visual field might follow the quantitative hypothesis, whereas others might follow a qualitative hypothesis for the same task.

To investigate the effect of M-scaling on contrast sensitivity and acuity across locations, we measured the whole contrast sensitivity function (CSF, known as the 'window of visibility') and manipulated stimulus eccentricity and size to assess how CS-peak (contrast sensitivity), SF-peak, SF-cutoff (acuity), and the area under the log CSF curve (AULCSF) vary across conditions and locations. Our main interest was to assess whether M-scaling, and thus cortical magnification, eliminates polar angle asymmetries in contrast sensitivity (*Figure 1B*).

To do so, we magnified stimulus size to equate the cortical representation for stimuli at different visual field locations (*Rovamo and Virsu, 1979*). By measuring contrast sensitivity of sinusoidal gratings at different regions of the visual field, *Rovamo and Virsu, 1979* derived a linear cortical magnification factor (CMF) that has been widely used. Linear cortical magnification (*M*) describes the distance along V1 corresponding to 1° of eccentricity and is expressed as millimeters of cortex per degree of visual angle. By applying this factor, one can equate the amount of cortex activated, regardless of retinal eccentricity, and achieve similar spatial and temporal CSFs. *Rovamo and Virsu, 1979* provided a specific M-scaling equation for each principal half meridian: nasal, temporal, superior, and inferior. This M-scaling procedure eliminates the eccentricity effect on performance in contrast sensitivity along these four half meridians, and their calculations have been used in many other studies for which visual performance differences across locations are eliminated once stimuli have been magnified (*Carrasco et al., 1998*; *Carrasco and Frieder, 1997*; *Goolkasian, 1994*; *Himmelberg and Wade, 2019*; *Prince and Rogers, 1998*; *Virsu et al., 1982*).

Here, observers performed an orientation discrimination task, which is contingent upon contrast sensitivity (*Nachmias, 1967*; *Olzak and Thomas, 2003*; *Pestilli et al., 2009*), when gratings appeared along the horizontal and vertical meridians, at 2° and 6° eccentricity. For the M-scale condition, gratings appeared at 6° eccentricity along horizontal and vertical meridians and the grating sizes were scaled separately for each polar angle meridian, based on meridian-dependent M-scaling equations (*Rovamo and Virsu, 1979*).

Surprisingly, cortically magnifying the stimuli to account for different cortical representations at the polar angle meridians did not eliminate polar angle asymmetries in contrast sensitivity, supporting the qualitative hypothesis. In contrast, and as expected, contrast sensitivity differences for eccentricity were eliminated, supporting the quantitative hypothesis. These differential results indicate that limits in contrast sensitivity as a function of eccentricity and polar angle likely emerge from different anatomical and computational constraints.

## Results

The CSF characterizes stimulus visibility. We measured human CSFs within the parafovea (2° eccentricity) and perifovea (6° eccentricity) at three polar angles: horizontal meridian (HM), lower vertical meridian (LVM), and upper vertical meridian (UVM). While maintaining fixation, observers reported the orientation of a target grating for which contrast and spatial frequency (SF) varied on each trial (*Figure 2A*). Using a parametric contrast sensitivity model, we characterized observers' CSFs along the HM and VM, before and after M-scaling (*Figure 2A and B*, see 'Methods').

We used model comparisons among nine CSF functional forms (*Chung and Legge, 2016*; *Movshon and Kiorpes, 1988*; *Watson and Ahumada, 2005*) to assess the differences across eccentricity and around polar angle for fixed-size gratings (*Figure 3*). The models were applied to group-level data. We extracted key CSF attributes – the peak contrast sensitivity (peak-CS), acuity limit (cutoff-SF), and

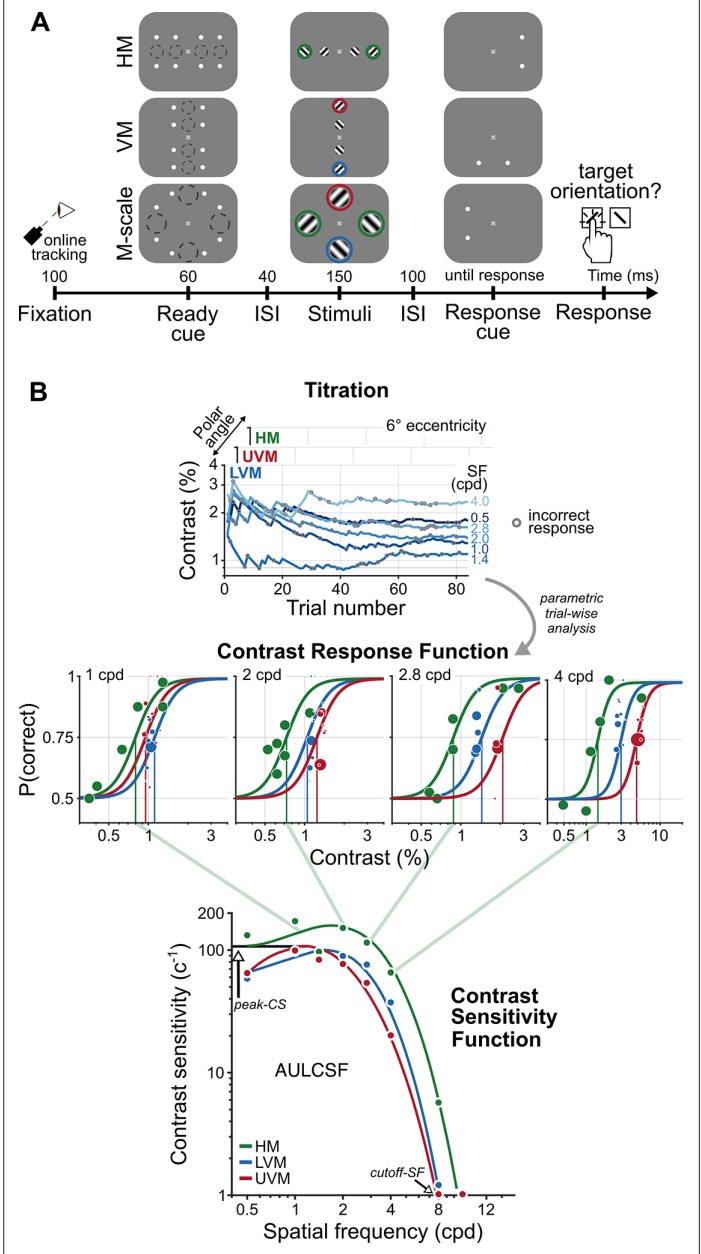

**Figure 2.** A psychophysical procedure to measure and a parametric model to characterize contrast sensitivity functions (CSFs). (**A**) An example trial sequence for the orientation discrimination task. Each trial began with a fixation period, after which a cue indicated the onset of four gratings. The dashed circles illustrate the location and size of the grating stimuli; they did not appear during the experiment. Gratings appeared in the parafovea (2° eccentricity) and perifovea (6° eccentricity), separately along the horizontal (HM) or vertical meridian (VM) or were M-scaled, and presented simultaneously at each meridional location in the perifovea (M-scale). A response cue indicated which grating observers should report. The colored circles indicate the perifoveal locations we compared to assess the impact of M-scaling on polar angle asymmetries: Green, HM; blue, lower VM (LVM); red, upper VM (UVM). (**B**) Parametric contrast sensitivity model. Grating contrast varied throughout the experiment following independent titration procedures for each eccentricity and polar angle location. Gray circles indicate incorrect responses for a given trial (top row). A model composed of contrast response functions (CRF, middle row) and CSFs (bottom row) constrained the relation between trial-wise performance, SF, eccentricity, and polar angle. The diagonal green lines depict the connection between contrast thresholds from individual CRFs to contrast sensitivity on the CSF for the HM; contrast sensitivity is the inverse of contrast threshold. The colored dots in each CRF and CSF depict a representative observer's task performance and contrast sensitivity, determined directly from the titration procedures. The colored lines depict the best-fitting model estimates. We derived key attributes of the

*Figure 2 continued on next page*

*Figure 2 continued*

CSF – peak contrast sensitivity (peak-CS), the acuity limit (cutoff-SF), and the area under the log contrast sensitivity function (AULCSF) – from the fitted parametric model.

area under the log contrast sensitivity function (AULCSF) – to characterize how contrast sensitivity changes with eccentricity, polar angle, and after M-scaling.

We magnified perifoveal gratings (6° eccentricity) following anisotropic M-scaling (*Rovamo and Virsu, 1979*) to equate their cortical representation with parafoveal (2° eccentricity) gratings (HM: 7.08°; LVM: 7.68°; UVM: 7.70°) and compared how this M-scaling changes CSFs attributes.

Contrast sensitivity peaked at a given SF and declined more rapidly for higher than lower SFs. We averaged CSFs across polar angle to isolate the eccentricity effect at 2°, 6°, and after M-scaling perifoveal CSFs (6°$_{M-scale}$; *Figure 4A*). CSFs decreased with eccentricity but less so after M-scaling. The eccentricity effect for fixed-size gratings, quantified as the percent change in contrast sensitivity for 2°, 6°, and 6°$_{M-scale}$, increased from ~30% to 120% across SF (*Figure 4B*).

After M-scaling, the eccentricity effect became negative for SFs <2 cpd, with higher contrast sensitivity at 6°$_{M-scale}$ than 2° (*Figure 4B*). A repeated-measures ANOVA (SF: 0.5–11 cpd; stimulus size: fixed vs. M-scaled) showed that M-scaling diminished the eccentricity effect differentially across SF (*interaction:* F(7,63) = 61.13, p<0.001, $\eta_G^2$ = 0.872). Post hoc *t*-tests revealed significant reductions in contrast sensitivity for all SFs (0.5, 1, 1.4, 2, and 2.8 cpd, p<0.001; 4, cpd, p<0.01) except the two highest SFs (8 and 11 cpd, p>0.1), which reached the acuity limit. Thus, consistent with previous results (*Rovamo et al., 1978*; *Rovamo and Virsu, 1979*), M-scaling slightly reversed typical eccentricity effects for low SFs and reduced them for medium SFs (*Figure 4—figure supplement 1*).

Contrast sensitivity across SFs was greater for the HM than VM at 2°, 6°, and 6°$_{M-scale}$ (*Figure 4C*). We quantified the HVA extent as the percent change in contrast sensitivity between the HM and VM (averaged LVM and UVM); positive values indicate higher sensitivity for the HM than VM. At 2° and 6°, the HVA extent increased from 20% to 120% across SF (*Figure 4D*). Remarkably, this HVA extent matched the eccentricity effect at high SFs (*Figure 4B*). Thus, differences in contrast sensitivity between the HM and VM at a fixed eccentricity were as pronounced as tripling stimulus eccentricity from 2° to 6°.

The HVA remained after M-scaling. A two-way ANOVA compared its extent at the perifovea (6° eccentricity) before and after M-scaling. M-scaling the stimulus reduced the HVA extent as a function of SF (*interaction:* F(7,63) = 7.32, p=0.0035, $\eta_G^2$ = 0.449). For all but one SF (8 cpd: p=0.021, 95% CI = [1.26 57.37], d = 0.75), M-scaling did not affect the HVA (p>0.05). This finding supports the 'qualitative' hypothesis – unlike eccentricity, the HVA must be mediated by factors beyond cortical

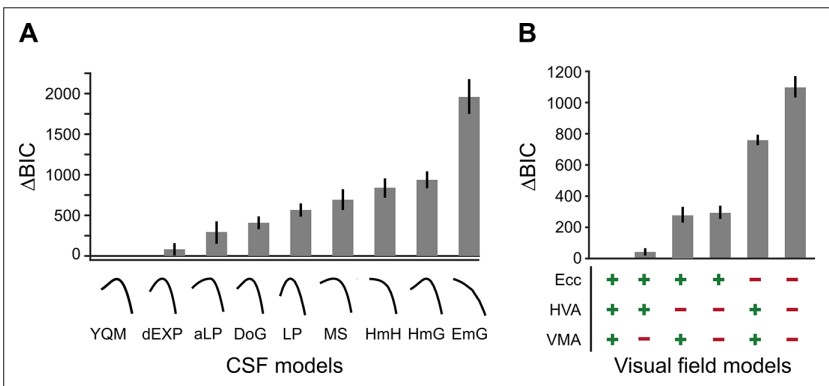

**Figure 3.** BIC model comparisons for contrast sensitivity function (CSF) and visual field models. (**A**) CSF model comparisons for the nine candidate functional forms of the CSF applied to the group data (*Table 1*). Low ΔBIC values indicate superior model performance. Curves under each bar illustrate the best fit of each CSF model to a representative observer (n=10). (**B**) Visual field model comparisons (*Table 2*). '+' and '-' under each bar indicate the components included and excluded, respectively, in each model. For example, '+' for 'HVA' indicates that CSFs could change between the horizontal and vertical meridians, whereas a '-' indicates that CSFs for the horizontal meridian were identical to the lower vertical meridian.

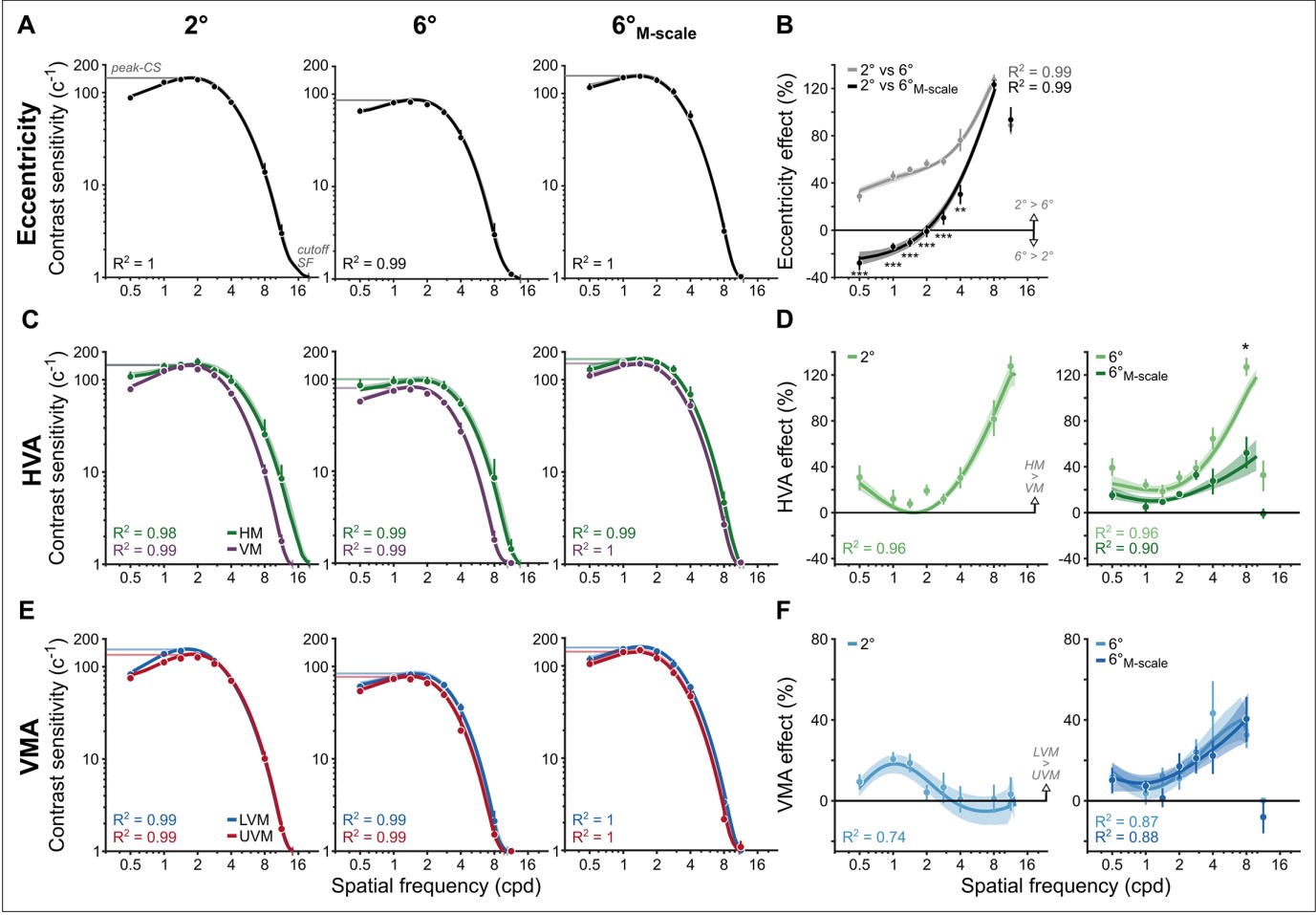

**Figure 4.** M-scaling diminishes the eccentricity effect, but neither the horizontal-vertical anisotropy (HVA) nor the vertical meridian asymmetry (VMA) (n=10). (**A**) Contrast sensitivity functions (CSFs) averaged across polar angles for fixed-size gratings at 2° and 6°, as well as for M-scaled gratings at 6°. (**B**) Eccentricity effects are quantified as the percent change in contrast sensitivity between 2° and 6° as well as between 2° and 6°$_{M-scale}$. Positive values indicate higher contrast sensitivity at 2° than 6°. Negative values indicate a reversal: higher contrast sensitivity at 6° than 2°. (**C**) CSFs for the horizontal meridian (HM) compared to the average CSF across the lower vertical meridian (LVM) and upper vertical meridian (UVM). (**D**) The percent change between horizontal and vertical meridians at 2° (left) and the percentage change between meridians for 6° and 6°$_{M-scale}$ (right). Values above 0% indicate higher sensitivity for the HM than VM. (**E**) CSFs for the LVM and UVM. (**F**) The percent change between LVM and UVM following the conventions in (**D**) (with a truncated y-axis); positive values indicate higher sensitivity for the LVM than UVM. All dots correspond to the group-average (n=10) contrast sensitivity and percent change in contrast sensitivity, as estimated from the titration procedures. Lines in panels (**A, C, E**) correspond to the group-average fit of the parametric contrast sensitivity model. Lines in panels (**B, D, F**) correspond to group average location percent differences as calculated in *Equation 9* (see 'Methods'). Note that the line does not reach the highest SF in these panels for the 6° and 6°$_{M-scale}$ comparison, as observers performed at chance, consistent with the fact that SF is harder to discriminate in the periphery. Error bars and shaded areas denote bootstrapped 68% confidence intervals. Repeated-measures ANOVA; *p<0.05, **p<0.01, ***p<0.001.

The online version of this article includes the following figure supplement(s) for figure 4:

**Figure supplement 1.** Qualitatively similar contrast sensitivity functions (CSFs) between previous reports (*Rovamo and Virsu, 1979*) and this study for fixed-size and M-scaled grating stimuli.

magnification, such as qualitatively different image-processing capabilities and/or neural computations around polar angle (*Figure 1B*).

Contrast sensitivity across SFs was higher along the LVM than UVM for 2°, 6°, and 6°$_{M-scale}$ (*Figure 4E*). We quantified the VMA extent as the percent change in contrast sensitivity between the LVM and UVM (*Figure 4F*). For fixed-size and M-scaled gratings, the VMA extent reached a maximum of 40% at 1 cpd in the parafovea and 8 cpd in the perifovea. The VMA has only been characterized at eccentricities > 2° (*Abrams et al., 2012*; *Cameron et al., 2002*; *Carrasco et al., 2001*; *Himmelberg et al., 2020*). This near-foveal location reveals that the SF at which the VMA peaks depends on eccentricity.

The VMA also remained after M-scaling (*Figure 4F*). A two-way ANOVA found a main effect of SF (F(7,63) = 10.16, p<0.001, $\eta_G^2$ = 0.53) due to an increasing perifoveal VMA extent across SF. We found neither a main effect of stimulus size nor an interaction effect (p>0.1), indicating no difference in VMA extent before and after M-scaling stimulus size. This finding further supports the 'qualitative' hypothesis – unlike eccentricity, the VMA must be mediated by factors beyond cortical magnification, such distinct neural computations and image-processing capabilities at the UVM and LVM (*Figure 1B*).

Key CSF attributes – peak-CS, cutoff-SF, and AULCSF – displayed changes consonant with eccentricity effects and polar angle asymmetries (*Figure 5*), but peak-SF and SF-bandwidth did not (*Figure 5—figure supplement 1*). We assessed each attribute with separate repeated-measures ANOVAs for the HVA and VMA across eccentricity and polar angle.

The HVA emerged in the peak-CS only in the perifovea (*interaction:* F(2,18) = 18.33, p<0.001, $\eta_G^2$ = 0.671; *Figure 5A*). Peak-CS fell between 2° and 6° (*HVA*: p<0.001, 95% CI = [0.404 0.555], d = 4.52; *VMA*: p<0.001, 95% CI = [0.513 0.651], d = 6.039), increased in the perifovea after M-scaling (*HVA*: p<0.001, 95% CI = [-0.639–0.483], d = −5.14; *VMA*: p<0.001, 95% CI = [-0.681–0.562], d = −7.50), and did not differ between 2° and 6°$_{M-scale}$ (p>0.1). Importantly, differences between HM and VM only emerged at 6° (p<0.001, 95% CI = [0.153 0.318], d = 2.04) and 6°$_{M-scale}$ (p<0.01, 95% CI = [0.054 0.175], d = 1.35). In contrast, the VMA emerged at 2°, 6°, and 6°$_{M-scale}$ (*polar angle main effect*: F(1,9) = 8.65, p<0.02, $\eta_G^2$ = 0.490; *Figure 5B*). These findings show that the HVA and VMA emerged in peak-CS, but the HVA only in the perifovea, whereas the VMA emerged at both eccentricities. Moreover, although M-scaling matched the peak-CS between the parafovea and perifovea, it did not equate contrast sensitivity around polar angle.

The HVA and VMA also emerged in the cutoff-SF, consistent with previous studies (*Barbot et al., 2021*; *Wilkinson et al., 2016*). M-scaling reduced the HVA extent (*interaction:* F(2,18) = 19.20, p<0.001, $\eta_G^2$ = 0.681; *Figure 5C*). The cutoff-SF decreased between HM and VM at 2° (p<0.001, 95% CI = [0.227 0.403], d = 2.56), 6° (p<0.001, 95% CI = [0.231 0.324], d = 4.24), and slightly less so at 6°$_{M-scale}$ (p=0.0326, 95% CI = [0.0251 0.146], d = 1.01). Thus, M-scaling did not eliminate either the HVA or the eccentricity effect in the cutoff-SF; it was smaller at 6° (p<0.001, 95% CI = [0.318 0.409], d = 5.73) and 6°$_{M-scale}$ (p<0.001, 95% CI = [0.324 0.431], d = 5.07) than at 2°.

The VMA extent in cutoff-SF only emerged in the perifovea (*interaction:* F(2,18) = 5.26, p=0.029, $\eta_G^2$ = 0.369; *Figure 5D*). It decreased between the LVM and UVM at 6° (p=0.0397, 95% CI = [0.0185 0.121], d = 0.972) and 6°$_{M-scale}$ (p=0.0481, 95% CI = [0.0163 0.123], d = 0.935). Therefore, M-scaling did not eliminate either the VMA or the eccentricity effect in cutoff-SF (*2° > 6°*: p<0.001, 95% CI = [0.297 0.391], d = 5.25; *2° > 6°$_{M-scale}$*: p<0.001, 95% CI = [0.235 0.291], d = 6.73). However, it increased the perifoveal cutoff-SF along the VM (*6°$_{M-scale}$ > 6°*: p<0.005, 95% CI = [-0.119 –0.0437], d = −1.54). In short, the HVA in cutoff-SF occurred at both eccentricities but only in the perifovea for the VMA. Critically, M-scaling did not equate cutoff-SF among polar angles.

Similar to cut-off SF, the HVA in AULCSF was evident in both the peri- and parafovea (*interaction*: F(2,18) = 17.98, p<0.001, $\eta_G^2$ = 0.667; *Figure 5E*). AULCSF was greater for the HM than VM (HVA) for 2° (p<0.001, 95% CI = [3.866 7.540], d = 2.22), 6° (p<0.001, 95% CI = [2.956 4.792], d = 3.02), and 6°$_{M-scale}$ (p<0.05, 95% CI = [0.569 2.314], d = 1.18). The AULCSF for HVA decreased between 2° and 6° (*HVA*: p<0.001, 95% CI = [5.440 6.988], d = 5.71) and between 2° and 6°$_{M-scale}$ (p<0.001, 95% CI = [4.249 6.803], d = 3.096). M-scaling did not eliminate the AULCSF HVA (p=0.397, 95% CI = [–1.630 0.252], d = −0.523).

The VMA in AULCSF was only evident in the perifovea (*interaction*: F(2,18) = 7.27, p<0.005, $\eta_G^2$ = 0.447; *Figure 5F*). AULCSF was greater for the LVM than UVM at 6° (p=0.007, 95% CI = [0.479 1.634], d = 1.309) and at 6°$_{M-scale}$ (p=0.010, 95% CI = [0.481 1.772], d = 1.248), but not at 2° (p=2.926, 95% CI = [–0.747 0.727], d = −0.01). The VMA extent decreased from 2° to 6° (p<0.001, 95% CI = [4.658 5.942], d = 5.91), from 2° to 6°$_{M-scale}$ (p<0.001, 95% CI = [2.942 3.849], d = 5.36), and between 6° and 6°$_{M-scale}$ (p<0.001, 95% CI = [-2.351–1.459], d = −3.05). Thus, M-scaling did not eliminate the VMA for AULCSF.

Next, we quantified the magnitude of the HVA and VMA for the 2°, 6°$_{M-scale}$ peak-CS, cutoff-SF, and AULCSF measurements. The HVA magnitude was calculated as the percent increase from the VM to HM, whereas the VMA magnitude was calculated as the percent increase from the LVM to UVM. We ran a series of one-way ANOVAs, and when appropriate, used post hoc *t*-tests to assess how the HVA and VMA magnitudes changed between 2° and 6°, and 6° and 6°$_{M-scale}$ conditions.

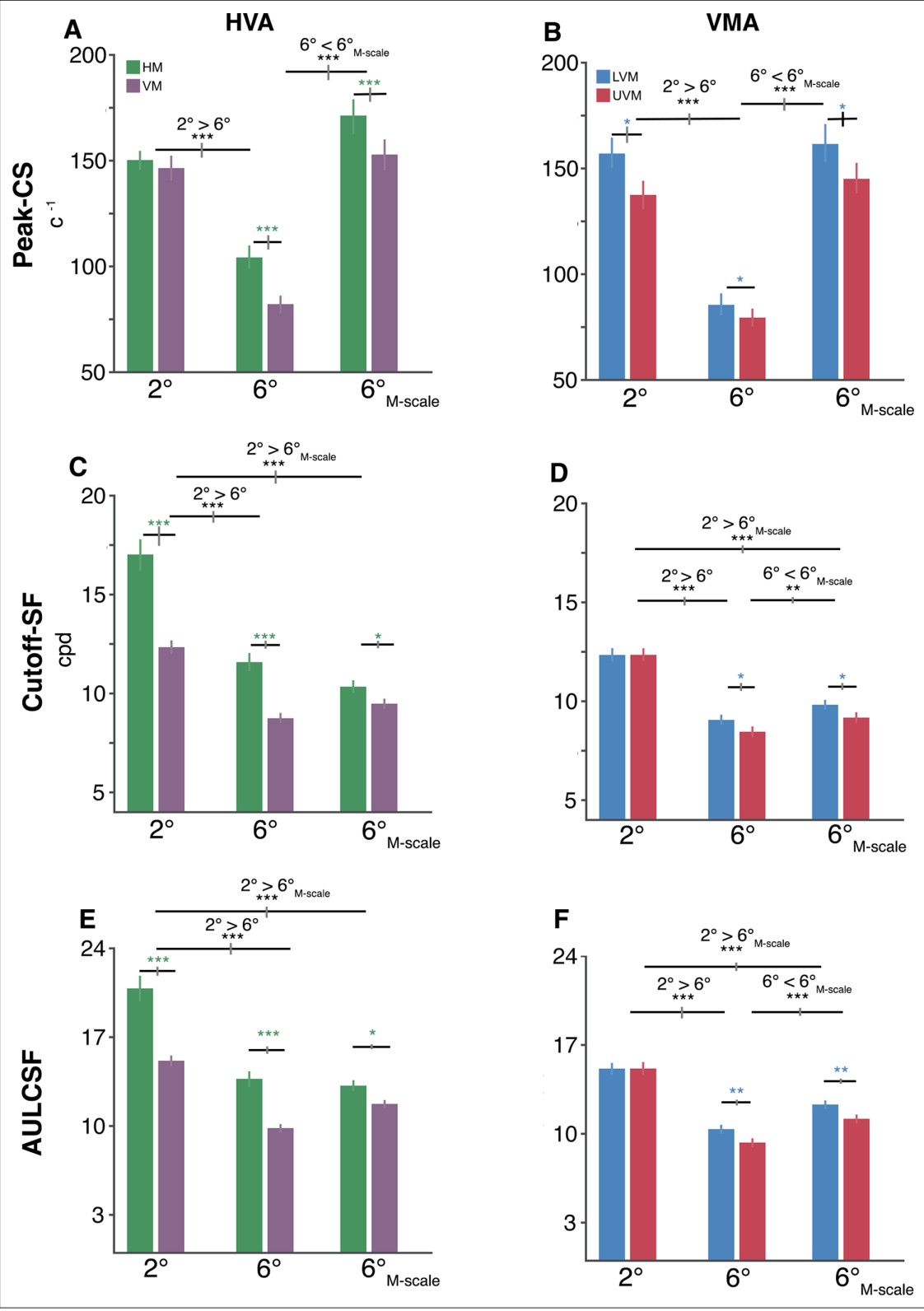

**Figure 5.** Polar angle asymmetries emerge in key contrast sensitivity function (CSF) attributes. (**A, B**) Peak contrast sensitivity for the horizontal-vertical anisotropy (HVA) and vertical meridian asymmetry (VMA), respectively. (**C, D**) Cutoff-SF for the HVA and VMA, respectively. (**E, F**) Area under the log contrast sensitivity function (AULCSF) for the HVA and VMA, respectively. Each bar depicts the group-average (n=10) attribute at a given location, and error bars depict bootstrapped 68% confidence intervals. Horizontal gray lines denote significant comparisons of an ANOVA and of post hoc

*Figure 5 continued on next page*

*Figure 5 continued*

comparisons. The vertical lines displayed on the gray bars depict the 68% confidence interval for the differences between eccentricities or locations. Repeated-measures ANOVA; *p<0.05, **p<0.01, ***p<0.001.

The online version of this article includes the following figure supplement(s) for figure 5:

**Figure supplement 1.** Neither polar angle asymmetries nor eccentricity effects emerge in retinal peak spatial frequency (SF) and SF bandwidth.

For peak-CS, the HVA magnitude (*ANOVA:* $F_{(2,27)}$ = 13.76, p<0.005) increased between 2° and 6° (p<0.001, CI = [-24.120 –11.691]). M-scaling the stimulus reduced, but did not eliminate the HVA magnitude (p=0.016, CI = [2.343 17.407]; *Figure 6A*). The VMA magnitude (*ANOVA:* $F_{(2,27)}$ = 0.51, p=0.6047) did not change between 2° and 6°, nor after M-scaling (*Figure 6B*).

For cutoff-SF, the HVA magnitude (*ANOVA:* $F_{(2,27)}$ = 17.44, p<0.001) did not change between 2° and 6° (p=0.230, CI = [–1.904 6.927]), and although M-scaling reduced the HVA magnitude (p<0.001, CI = [8.661 23.721]), the HVA was still evident in the 6°$_{M-scale}$ condition. The VMA magnitude (*ANOVA:* $F_{(2,27)}$ = 3.93, p=0.031) increased between 2° and 6° (p=0.028, CI = [-12.472 –0.902]), and M-scaling did not alter the VMA magnitude at 6° (p=0.985, CI = [–5.844 5.948]).

For AULCSF, the HVA magnitude (*ANOVA:* $F_{(2,27)}$ = 12.82, p<0.001) did not change between 2° and 6° (p=0.443, CI = [–5.605 2.671]), and although M-scaling reduced the HVA magnitude (p<0.001, CI = [9.181 25.393]), the HVA was still evident in the 6°$_{M-scale}$ condition. The VMA magnitude (*ANOVA:* $F_{(2,27)}$ = 6.25, p=0.0177) increased between 2° and 6° (p=0.009, CI = [-18.323 –2.257]), but M-scaling did not alter the VMA magnitude at 6° (p=0.682, CI = [–4.615 6.732]).

These data show that M-scaling stimulus size based on its cortical representation eliminates differences in contrast sensitivity as a function of eccentricity, but not polar angle. fMRI work shows that there is an HVA and VMA in V1 surface area (*Benson et al., 2021*; *Himmelberg et al., 2023a*; *Himmelberg et al., 2022b*, *Himmelberg et al., 2021*; *Silva et al., 2018*), and that individual differences in these cortical asymmetries correlate with contrast sensitivity measurements (*Himmelberg et al., 2022b*). Here, we measured the distribution of V1 surface area at the polar angle meridians and confirmed that individual measurements of V1 surface area correlate with contrast sensitivity across our observers. We correlated the amount of V1 surface area representing ±15° wedge-ROIs (1–8° of eccentricity) centered along the HM, UVM, and LVM of the visual field with the respective peak-CS measurement at 2°, 6°, and 6°$_{M-scale}$ for 7 of our 10 observers for whom we could obtain fMRI-derived retinotopic maps.

First, and in line with previous work, at the group level, there was more V1 surface area representing the HM than VM (p=0.001), and the LVM than UVM of the visual field (p=0.031) (*Figure 7A*). Next, we found that, across observers, V1 surface area measurements taken from a meridian correlated (one-tailed Spearman's correlations) with the contrast sensitivity measurements from the corresponding meridian for the 6° (r = 0.40, p=0.036; *Figure 7C*) and 6°$_{M-scale}$ (r = 0.39, p=0.040; *Figure 7D*) stimulus conditions, but not 2° (r = 0.16, p=0.400; *Figure 7B*). These positive correlations indicate that, for our observers, V1 surface area is linked to contrast sensitivity measurements, thus M-scaling *should* correct for polar angle differences in the cortical representation. However, correlating the difference in contrast sensitivity at each meridian, after M-scaling the stimulus size, against V1 surface area at the corresponding meridian yield a nonsignificant correlation (two-tailed Spearman's correlation; r = 0.20, p=0.393).The finding that M-scaling does not correct for the cortical representation at the HM, LVM, and UVM supports the 'qualitative hypothesis' – that there are additional underlying neural and computational factors beyond V1 cortical magnification that contribute to perceptual polar angle asymmetries.

## Discussion

We investigated whether the quantitative or qualitative hypothesis can explain the differences in contrast sensitivity and acuity across eccentricity and around polar angle in the visual field. We found that M-scaling stimulus size, to equate for the differences of cortical representation as function of eccentricity and polar angle, eliminated the differences in contrast sensitivity as a function of eccentricity, in line with the quantitative hypothesis, but not polar angle, in line with the qualitative hypothesis.

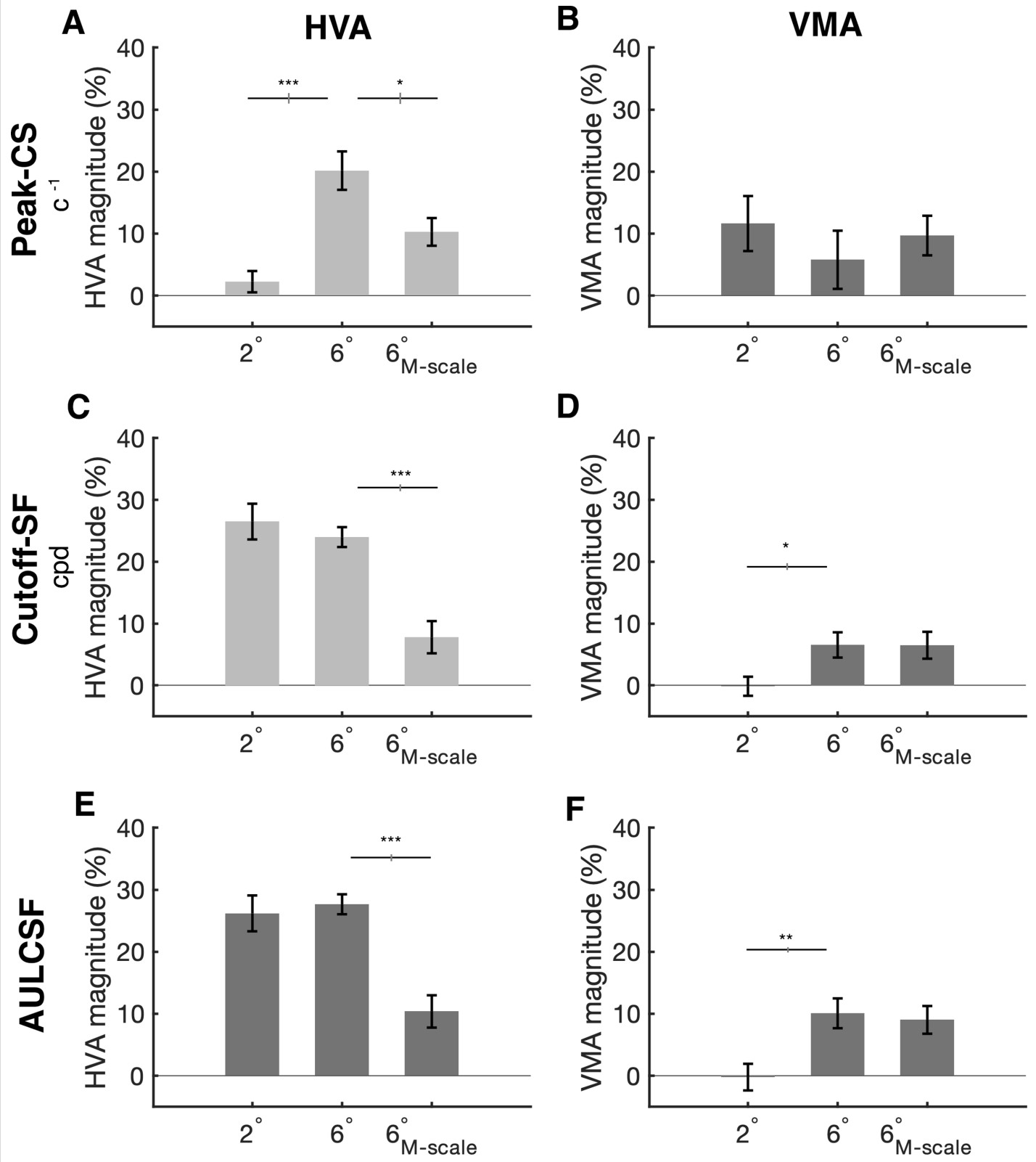

**Figure 6.** Horizontal-vertical anisotropy (HVA) and vertical meridian asymmetry (VMA) magnitudes for peak contrast sensitivity, spatial frequency cutoff, and area under the log contrast sensitivity function curve (AULCSF). (**A, B**) Peak contrast sensitivity for the HVA and VMA, respectively, at 2°, 6°, and 6°$_{M-scale}$. (**C, D**) Cutoff-SF for the HVA and VMA, respectively, at 2°, 6°, and 6°$_{M-scale}$. (**E, F**) AULCSF for the HVA and VMA, respectively, at 2°, 6°, and 6° M-scaled. n=10. Error bars representing ±1 standard error of the mean (SEM) and horizontal gray lines denote significant comparisons of an ANOVA and post hoc comparisons. Vertical lines displayed on the gray bars denote the standard error of the difference (SED). *p<0.05, **p<0.01, ***p<0.001.

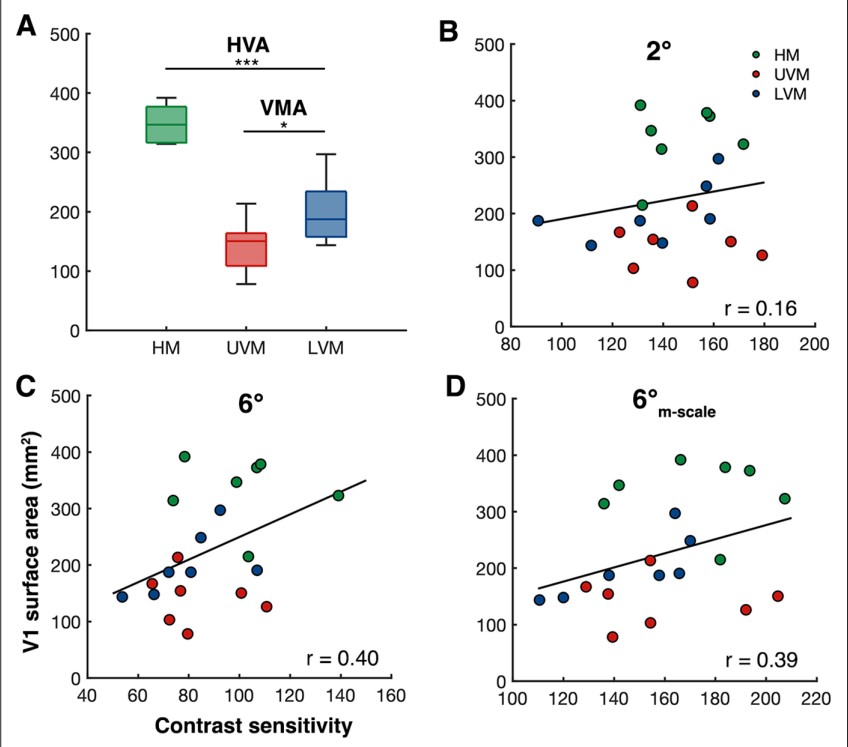

**Figure 7.** Individualized V1 surface area measurements at the cardinal meridians correlate with peak contrast sensitivity measurements. (**A**) Group-level V1 surface area measurements (n=7) taken from the cortical representation of the horizontal meridian (HM) (mean of left and right HM), the upper vertical meridian (UVM), and lower vertical meridian (LVM). *p<0.05, ***p<0.001. (**B–, C**) Between-subject Spearman's correlations of V1 surface area (±15° of angle, 1–8° eccentricity) at each meridian with peak contrast sensitivity measurements (n = 7, 3 measurements per observer) at the same meridian for (**B**) 2°, (**C**) 6°, and (**D**) 6°-M-scale stimulus conditions.

## M-scaling eliminates differences in contrast sensitivity as a function of eccentricity

Converging neural evidence demonstrates that cortical magnification limits peripheral vision. V1 surface area across eccentricity correlates with various perceptual measures, including acuity (*Duncan and Boynton, 2003*; *Song et al., 2015*), perceived angular size (*Murray et al., 2006*), and perceived object size (*Schwarzkopf et al., 2011*; *Schwarzkopf and Rees, 2013*). These perceptual differences across eccentricity arise from quantitative differences in the number of neurons for foveal and peripheral eccentricities. Consequently, accounting for cortical magnification via M-scaling diminishes or eliminates eccentricity effects. Our present results support these findings; M-scaling stimulus size diminished the difference in contrast sensitivity and acuity between 2° and 6°, indicating that cortical magnification predominantly underlies performance differences as a function of eccentricity.

## M-scaling does not eliminate differences in contrast sensitivity as a function of polar angle

In contrast to the effect on eccentricity, M-scaling stimulus size did not eliminate differences in contrast sensitivity as a function of polar angle. After M-scaling stimulus size based on the meridian-dependent functions provided by *Rovamo and Virsu, 1979*, the HVA and VMA remained.

The finding that M-scaling does not eliminate polar angle asymmetries for contrast sensitivity and acuity is surprising as perceptual polar angle asymmetries have been linked to V1 cortical magnification. First, psychophysical measures of the HVA and VMA magnitude for contrast sensitivity (*Abrams et al., 2012*; *Himmelberg et al., 2022b*; *Himmelberg et al., 2020*) and acuity (*Barbot et al., 2021*; *Benson et al., 2021*) provide a close match with the cortical HVA and VMA; there is ~60% more V1 tissue representing the HM than VM, and ~25% more representing the LVM than UVM (*Benson et al.,*

2021; *Himmelberg et al., 2021*; *Himmelberg et al., 2023a*; *Himmelberg et al., 2022b*). Thus, there are asymmetries in the distribution of V1 neurons that parallel behavior. Second, individual differences in contrast sensitivity at each of the cardinal meridians correlate with localized measures of the amount of V1 surface representing the same meridians (*Himmelberg et al., 2022b*). We found the same correlation for the contrast sensitivity measurements here, albeit with a reduced number of observers – which speaks to the high-level of reproducibility of location-specific brain–behavior correlations using retinotopic data (*Himmelberg et al., 2022a*). Thus, M-scaling stimulus size to compensate for cortical magnification around the polar angle *should,* in principle, equate contrast sensitivity. But here, we found it does not.

Our data showed that the magnitude of the HVA was larger than the VMA, consistent with prior work (*Himmelberg et al., 2020*). The magnitude of the HVA and VMA differed among the three stimulus conditions (2°, 6°, and 6°<sub>M-scale</sub>) for the key CSF properties: (1) peak-CS, (2) SF-cutoff, and (3) AULCSF. (1) For peak-CS, there was a relatively weak HVA at 2°. The HVA magnitude increased at 6° eccentricity, consistent with previous studies (*Baldwin et al., 2012*; *Carrasco et al., 2001*; *Fuller et al., 2008*; *Greenwood et al., 2017*), and was still evident after the M-scaling stimulus size. On the other hand, the VMA magnitude was consistent across at 2°, 6°, and after M-scaling. Thus, both the HVA and VMA remained after M-scaling for peak-CS. (2) For cutoff-SF, the data showed a large HVA at 2° and 6°. M-scaling stimulus size decreased the magnitude of the HVA, but nonetheless the HVA remained. Likewise, the VMA for cutoff-SF remained after M-scaling stimulus size. (3) The HVA and VMA magnitude for AULCSF mimicked the results found for cutoff-SF; the HVA magnitude was large for 2° and 6°, and was reduced – but still clear – after M-scaling. The VMA was evident at 6° and persisted after M-scaling. Overall, across our three key CSF parameters, M-scaling stimulus size decreased the magnitude of the HVA and VMA, but did not eliminate the perceptual asymmetries. Thus, the asymmetries persisted after equating for their cortical representation.

There was an apparent reduction in the HVA extent at a high SF (8 cpd, *Figure 4D*). This may have resulted from the fact that M-scaling the stimulus slightly *increased* cutoff-SF for the VM, consistent with the notion that it should reduce the effect of eccentricity, but slightly *decreased* the cutoff-SF for the HM, which was unexpected (*Figure 5C*). Indeed, we found that individual scores for these differences were marginally correlated ($r = 0.54$, p=0.056), suggesting that for the same observers for whom M-scaling reduced the detrimental effect of eccentricity more along the VM, surprisingly it had the opposite effect for the HM. We do not know the source of this effect.

Might M-scaling eliminate polar angle asymmetries for visual dimensions other than contrast sensitivity? Polar angle asymmetries have been identified for fundamental basic visual properties (e.g., contrast sensitivity; *Abrams et al., 2012*; *Baldwin et al., 2012*; *Cameron et al., 2002*; *Himmelberg et al., 2020*; *Pointer and Hess, 1989*; *Silva et al., 2008*; acuity; *Barbot et al., 2021*; *Greenwood et al., 2017*; *Kwak et al., 2023*; *Schwarzkopf, 2019*; *Wang et al., 2020*), for mid-level properties (e.g., crowding; *Greenwood et al., 2017*; *Petrov and Meleshkevich, 2011*) and texture segmentation (*Barbot et al., 2021*; *Greenwood et al., 2017*; *Kwak et al., 2023*; *Talgar and Carrasco, 2002*; *Wang et al., 2020*), and for higher-order properties (e.g., speed of information accrual; *Carrasco et al., 2004*; numerosity processing; *Chakravarthi et al., 2022*; face perception; *Afraz et al., 2010*; *Peterson and Eckstein, 2013*; and perceived object size; *Schwarzkopf, 2019*). However, contrast sensitivity is the currency of the visual system, which most – if not all – visual dimensions depend upon in some capacity. Thus, if M-scaling does not eliminate the polar angle asymmetries at the most fundamental level, then it is unlikely to eliminate asymmetries for higher-order dimensions – although this has yet to be empirically tested.

Together, these novel findings support the 'qualitative' hypothesis. M-scaling stimulus size based on the cortical representation as a function of polar angle diminished – but did not eliminate – the HVA, and had no effect on the VMA. These findings suggest that performance differences as a function of polar angle are likely to be mediated by both the cortical representation and by differential computations or image-processing capabilities of these neurons (i.e., differently tuned spatial filters).

We note that observers' viewing distance changed when the stimuli were presented at the HM and the VM for the non-scaled stimulus size. This change in distance changed monitor luminance (23 cd/m² and 19 cd/m², respectively). However, this change in luminance is not large enough to significantly affect contrast measurements (*Rahimi-Nasrabadi et al., 2021*). Moreover, these asymmetries in contrast sensitivity have been reported in many studies for which the viewing distance has been

constant (*Abrams et al., 2012*; *Baldwin et al., 2012*; *Barbot et al., 2021*; *Cameron et al., 2002*; *Carrasco et al., 2022*; *Carrasco et al., 2001*; *Hanning et al., 2022a*; *Himmelberg et al., 2022b*; *Himmelberg et al., 2020*; *Pointer and Hess, 1989*; *Regan and Beverley, 1983*; *Rijsdijk et al., 1980*; *Robson and Graham, 1981*; *Rosén et al., 2014*; *Silva et al., 2008*).

## What mechanism might underlie perceptual polar angle asymmetries?

If M-scaling stimulus size does not eliminate the polar angle asymmetries for contrast sensitivity, then what might be their underlying substrate? Perceptual asymmetries have been linked to the V1 properties (surface area, population receptive field [pRF] size, and BOLD amplitude) at the group- (*Benson et al., 2021*; *Himmelberg et al., 2021*; *Liu et al., 2006*; *Moutsiana et al., 2016*; *O'Connell et al., 2016*; *Silva et al., 2018*) and individual level (*Himmelberg et al., 2022b*). Further, here we show that individual differences in contrast sensitivity in our own data correlate with individual differences in V1 surface area along the polar angle meridians. Thus, the asymmetries must be explained by cortical magnification to some extent. What factors beyond neural count could contribute to the perceptual asymmetries, for which M-scaling cannot account for? To answer this, we are currently using reverse correlation to investigate whether and how eccentricity (*Xue et al., 2022*) and polar angle (*Xue and Carrasco, 2023*) alter orientation and SF tuning functions.

M-scaling has been shown to work for fundamental visual dimensions (*Cowey and Rolls, 1974*; *Di Russo et al., 2005*; *Himmelberg and Wade, 2019*; *Levi et al., 1999*; *Ludvigh, 1941*; *Rovamo et al., 1978*; *Virsu et al., 1982*; *Virsu and Rovamo, 1979*; *Wertheim, 1894*) – and fail for others, typically (but not always) more complex dimensions (*Hilz et al., 1981*; *Levi and Klein, 1986*; *Solomon and Sperling, 1995*; *Strasburger et al., 1994*; *Strasburger et al., 1991*; *Tyler, 1999*). Critically, here we found that for the *same* visual task and dimension, M-scaling stimulus size works for certain locations – eccentricity – but fails for others – cardinal polar angles. One possibility is that between-subject variability in the V1 polar angle representation underlies the inability of M-scaling to extinguish perceptual polar angle asymmetries. There is substantial variability in the size of V1 (*Benson et al., 2022*; *Dougherty et al., 2003*; *Himmelberg et al., 2022b*; *Moutsiana et al., 2016*) and how V1 tissue is distributed throughout the visual field (*Benson et al., 2022*; *Himmelberg et al., 2023a*; *Himmelberg et al., 2022b*). Another possibility is that there is greater variability in the cortical representation of polar angle compared to eccentricity that is not accounted for by the M-scaling equations that were derived from group-level data (*Rovamo and Virsu, 1979*). Finally, although we found a significant correlation between V1 surface area and contrast sensitivity at each meridian, we did not find a correlation between V1 surface area and the change in contrast sensitivity after M-scaling at each meridian. This suggests that even if stimulus size were adjusted via M-scaling equations based on individualized V1 surface measures, the perceptual asymmetries would likely remain.

## Conclusions

We used psychophysics to probe the neural substrates of contrast sensitivity across and around the visual field. We found striking polar angle asymmetries in contrast sensitivity, which were as pronounced as tripling eccentricity. The asymmetries were still present after M-scaling stimulus size. The M-scaling estimate provided by *Rovamo and Virsu, 1979* at the group level eliminated the decline in contrast sensitivity with eccentricity, but only diminished the HVA, and did not alter the VMA. These findings reveal that limits in contrast sensitivity across eccentricity and around polar angle likely emerge from different anatomical and computational constraints, and challenge the generalizability of the established view that cortical magnification limits basic visual perception throughout the visual field (*Duncan and Boynton, 2003*; *Rovamo and Virsu, 1979*; *Schwarzkopf et al., 2011*; *Schwarzkopf and Rees, 2013*; *Song et al., 2015*; *Virsu and Rovamo, 1979*). Although differences in contrast sensitivity at different eccentricities are predominantly mediated by cortical magnification, differences as a function of polar angle must be constrained by additional computational and neural image-processing capabilities. Models of spatial vision linking brain and behavior should account for what constrains basic visual perception not only across – but also around – the visual field.

## Methods

### Observers

We based our sample size on research on the impact of eccentricity (*Jigo and Carrasco, 2020*) and polar angle asymmetries on contrast sensitivity (*Cameron et al., 2002*) and acuity (*Barbot et al., 2021*). Ten observers with normal or corrected-to-normal vision participated in three conditions (eight females, aged 21–32 y, two authors: MJ and DT). All observers provided written informed consent under the University Committee's protocol on Activities Involving Human Subjects at New York University agreeing to participate in the study and the public release of their data. All experimental procedures were approved by the Ethics Committee at the NYU Department of Psychology (IRB: FY2016-466) and in agreement with the Declaration of Helsinki. All observers, except the authors, were naïve to the purpose of the study and were paid $12/hr. Data and code pertaining to the experiment are available on the OSF repository (https://osf.io/gvkdh/; *Jigo et al., 2023*).

### Stimuli

#### Gratings

Sinusoidal gratings with an SF of 0.5, 1, 1.4, 2, 2.8, 4, 8, or 11.3 cpd served as targets. For the HM condition, stimuli appeared along the left and right HM at 2° and 6° eccentricity. Similarly, stimuli appeared at the same eccentricities but along the upper and LVM for the VM condition. During the HM and VM conditions, a two-dimensional cosine function (4° wide, centered on the grating's peak luminance) windowed each grating at 2° and 6° eccentricity. For the M-scale condition, gratings appeared at 6° eccentricity along HM and VM. We scaled grating sizes separately for each polar angle, based on meridian-dependent M-scaling equations (*Rovamo and Virsu, 1979*), resulting in gratings that subtended 7.68° for the LVM, 7.70° for the UVM, and 7.08° for the HM.

Specifically, we computed M-scaled sizes as

$$S_{Xb} = \frac{S_{Xa} \times M_{Xa}}{M_{Xb}} \tag{1}$$

where $S_{Xb}$ corresponds to the magnified size in degrees of visual angle along meridian $X$ at eccentricity $b$. This M-scaled size equates the cortical representation with that of a grating of size $S_{Xa}$, which equaled 4°, positioned along the same meridian but at a different eccentricity $a$. $M_{Xa}$ and $M_{Xb}$ correspond to cortical magnification in mm/° along a given meridian $X$ at eccentricity $a$ and $b$, respectively.

Cortical magnification differed among meridians. For the LVM:

$$M_{LVM} = M_0 \left(1 + 0.42E + 0.000055E^3\right)^{-1} \tag{2}$$

where $M_0$ corresponds to cortical magnification at the central fovea, which was set to 7.99 mm/° and $E$ corresponds to the eccentricity of the stimulus.

Similarly, for the UVM:

$$M_{UVM} = M_0 \left(1 + 0.42E + 0.00012E^3\right)^{-1} \tag{3}$$

For the HM, we used the cortical magnification equations for both the nasal ($N$) and temporal ($T$) meridians:

$$M_N = M_0 \left(1 + 0.33E + 0.00007E^3\right)^{-1} \tag{4}$$

$$M_T = M_0 \left(1 + 0.29E + 0.000012E^3\right)^{-1} \tag{5}$$

and computed the M-scaled size at eccentricity $b$ for the HM ($S_{HM_b}$) using the average among M-scaled sizes for nasal ($S_{Nb}$) and temporal meridians ($S_{Tb}$):

$$S_{HM_b} = 0.5 \left(S_{Nb} + S_{Tb}\right) \tag{6}$$

### Cues

'Ready cues' prepared observers for the onset of the grating stimuli and 'response cues' indicated which grating to respond to. Response cues comprised a pair of white dots displaced 3.75° from the VM or HM for target gratings that appeared at those respective locations. Ready cues comprised the same white dots that appeared at all possible target locations for the HM (i.e., LHM and RHM) and VM (i.e., UVM and LVM) conditions.

### Fixation and background

Observers maintained their gaze on a gray fixation cross (17 cd/m$^2$) that subtended 0.35° and remained on the screen throughout the entire experiment. All stimuli appeared on a medium gray display background (26 cd/m$^2$).

## Apparatus

We generated visual stimuli on an Apple iMac using MGL (*Gardner et al., 2018*), a set of OpenGL libraries running in MATLAB (MathWorks, Natick, MA). Stimuli were displayed on a cathode ray tube (CRT) monitor (1280 × 960; 100 Hz). We gamma-corrected the monitor at a central location using a Konica Minolta LS-100 (Tokyo, Japan) with 8-bit accuracy. Observers sat in a dark and sound-proofed room and viewed the display binocularly with their heads stabilized by a chin-and-head rest positioned either 57 cm (VM and M-scale conditions) or 115 cm (HM condition, to display the highest SF tested, 16 cpd). The mean luminance of the display (from retina to monitor) was 23 cd/m$^2$ at 57 cm and 19 cd/m$^2$ at 115 cm. This difference in luminance does not significantly affect pupil size (<0.5 mm) or contrast sensitivity (*Rahimi-Nasrabadi et al., 2021*; *Rovamo et al., 1992*). An Eyelink 1000 eye tracker (S.R. Research, Ottawa, Ontario, Canada) monitored monocular eye position at 500 Hz (*Cornelissen et al., 2002*).

## Behavioral protocol

We instructed observers to maintain fixation. Stimulus presentation was contingent upon fixation on a central cross for 100 ms, after which the ready cue appeared (60 ms for the dots, 300 ms for the 'N'). The cue informed observers of the temporal onset of the target grating but provided no information about its location. Following an interstimulus interval (ISI; 40 ms for the dots, 100 ms for the 'N'), four gratings with the same SF appeared for 150 ms. Grating contrast varied for each trial, determined by independent adaptive titration procedures for each grating (see 'Titration'). A 100 ms ISI and the response cue followed the grating presentation. The response cue indicated which grating observers should report on each trial. Observers performed an orientation discrimination task. They used the right or left arrow keys on a keyboard to report whether the cued grating was tilted left or right of vertical. If the eye blinked or deviated >1° from fixation, the trial was immediately aborted and rerun at the end of the block.

Observers were instructed to be as accurate as possible, without time stress. They received auditory feedback for incorrect responses on a trial-by-trial basis. Once observers finished a block, the monitor displayed their overall accuracy (percent correct) as feedback.

## Procedure

Observers performed three conditions: HM, VM, and M-scale. For the HM condition, they completed 1080 trials per location (left and right HM; 160 trials per SF), for the VM condition 1344 per location (UVM and LVM; 140 trials per SF), and for the M-scale condition 1008 per location (84 trials per SF). On each trial, we randomly interleaved the target's orientation, SF, eccentricity, and/or polar angle (either left and right HM, or UVM and LVM), and adjusted grating contrast based on task performance (see 'Titration'). Before the main experimental sessions, observers completed a single practice block of trials to familiarize themselves with the stimuli and task.

## Titration

For VM and M-scale conditions, we titrated contrast separately for each combination of SF, eccentricity, and polar angle with best PEST, a maximum likelihood adaptive procedure, using custom code (https://github.com/michaeljigo/palamedes_wrapper; *Jigo, 2021*) that ran subroutines implemented

in the Palamedes toolbox (**Prins and Kingdom, 2018**). For HM, we used a 3-down, 1-up weighted staircase (**García-Pérez, 1998**). Both titration procedures targeted 75% task performance.

## Parametric contrast sensitivity model

We fit a parametric model that linked contrast response functions (CRFs) and CSFs to observers' binary decisions (CW vs. CCW) on individual trials. Our model includes (1) a logistic function for the CRF, with slope fixed across SF (**Jigo and Carrasco, 2020**) and asymptotes matched to the specifications of the adaptive titration procedure; (2) nine candidate models of the CSF; and (3) six visual field models that specify how contrast sensitivity changes with eccentricity and polar angle.

### CRF

We characterized the CRF – performance as a function of $\log_{10}$-transformed contrast – using a logistic function (**Equation 7**) with lower and upper asymptotes ($p_l = 0.5$, $p_u = 0.99$) and slope ($\kappa = 11.8$) matching the specifications of the adaptive titration procedure, as well as a $\log_{10}$-transformed contrast threshold ($c_t(f, r, \theta)$, **Equation 8**) that targets 75% discrimination accuracy ($p_t = 0.75$) at each SF ($f$), eccentricity ($r$) and polar angle ($\theta$):

$$p(c, f, r, \theta) = s\left(\frac{1}{1 + exp(-\kappa(c - t))}\right) + p_l \tag{7}$$

**Table 1.** Candidate contrast sensitivity function (CSF) models.
The number of parameters included in each model ($n$) is denoted under the corresponding label, along with a brief description and the model equation. The bolded entry indicates the best-fitting model.

| Label ($n$) | Description | Equation |
|---|---|---|
| YQM (4) | Derived from a model of contrast sensitivity **Chung and Legge, 2016** | $c_t^{-1}(f, r, \theta; \alpha, \beta, \gamma, \delta) = \delta\left[\dfrac{exp(-f/\alpha)}{1 + \frac{\gamma}{1 + (f/\beta)}}\right]$ |
| dEXP (3) | Double exponential function **Chakravarthi et al., 2022** | $c_t^{-1}(f, r, \theta; \alpha, \beta, \gamma, \delta) = \delta f^\alpha exp(-f/\beta)$ |
| aLP (4) | Asymmetric log parabola **Corbett and Carrasco, 2011** | $c_t^{-1}(f, r, \theta; \alpha, \beta, \gamma, \delta) = \left\{\delta(f - a)^2 \beta^2 \text{ if } f < \alpha\right.$ <br> $c_t^{-1}(f, r, \theta; \alpha, \beta, \gamma, \delta) = \left\{\delta(f - a)^2 \gamma^2 \text{ if } f \geq \alpha\right.$ |
| DoG (4) | Difference of Gaussians **Chung and Legge, 2016** | $c_t^{-1}(f, r, \theta; \alpha, \beta, \gamma, \delta) = \delta\left(exp\left[-(f/\alpha)^2\right] - \gamma exp\left[-(f/\beta)^2\right]\right)$ |
| LP (3) | Log parabola **Chung and Legge, 2016** | $c_t^{-1}(f, r, \theta; \alpha, \beta, \gamma, \delta) = \delta exp\left[-\left(\frac{\log_2\left(\frac{f}{\alpha}\right)}{\beta}\right)^2\right]$ |
| MS (4) | Generalized Gaussian with linear function of SF **Chung and Legge, 2016** | $c_t^{-1}(f, r, \theta; \alpha, \beta, \gamma, \delta) = \delta\left(1 - \beta + \frac{f}{\alpha}\right) exp\left[-(f/\alpha)^\gamma\right]$ |
| HmH (4) | Difference of hyperbolic secants **Chung and Legge, 2016** | $c_t^{-1}(f, r, \theta; \alpha, \beta, \gamma, \delta) = \delta\left(sech\left[f/\alpha\right] - \gamma sech\left[f/\beta\right]\right)$ |
| HmG (4) | Hyperbolic secant minus a Gaussian **Chung and Legge, 2016** | $c_t^{-1}(f, r, \theta; \alpha, \beta, \gamma, \delta) = \delta\left(sech\left[f/\alpha\right] - \gamma exp\left[-(f/\beta)^2\right]\right)$ |
| EmG (4) | Exponential minus a Gaussian **Chung and Legge, 2016** | $c_t^{-1}(f, r, \theta; \alpha, \beta, \gamma, \delta) = \delta\left(exp\left[-f/\alpha\right] - \gamma exp\left[-(f/\beta)^2\right]\right)$ |

**Table 2.** Models of contrast sensitivity across eccentricity and polar angle.
The bolded entry indicates the best-fitting model.

| Model label | Description | Max number of parameters |
|---|---|---|
| **Ecc + HVA + VMA** | **CSFs vary across eccentricity and polar angle** | **24** |
| Ecc +HVA - VMA | CSFs do not vary along the VM | 16 |
| Ecc - HVA + VMA | CSFs do not vary along the HM | 16 |
| -Ecc + HVA + VMA | CSFs do not vary across eccentricity | 12 |
| Ecc - HVA - VMA | CSFs do not vary along the VM and HM | 8 |
| -Ecc - HVA - VMA | CSFs are identical at all visual field locations | 4 |

CSF: contrast sensitivity function; VM: vertical meridian; HM: horizontal meridian; HVA: horizontal-vertical anisotropy; VMA: vertical meridian asymmetry.

where $s = 1 - (1 - p_u) - p_l$ , which scales the dynamic range of the function. Because contrast was $\log_{10}$-transformed, adjusting the contrast threshold in *Equation 7* yields rigid shifts in logarithmic contrast.

In *Equation 7*, $t$ corresponds to a transformation of contrast threshold, which ensures $p_t$ is accurately targeted given the constraints of the logistic function's slope, upper and lower asymptotes:

$$t\left(c_t(f, r, \theta), p_r\right) = c_t(f, r, \theta) - \kappa^{-1} log\left(\frac{p_r}{1-p_r}\right) \tag{8}$$

where $p_r$ denotes the ratio between the targeted performance level and the dynamic range of the CRF: $p_r = \frac{p_t - p_l}{s}$ .

## CSF

Contrast sensitivity typically peaks at a given SF and declines precipitously for higher SFs and gradually for lower SFs (*Chakravarthi et al., 2022*; *Chung and Legge, 2016*; *Kelly, 1977*; *Corbett and Carrasco, 2011*). We implemented this pattern by constraining the contrast threshold ($c_t (f, r, \theta)$ in *Equation 8*) across SF to adhere to a functional form of the CSF. We implemented nine candidate CSF models that each determined contrast sensitivity ($c_t^{-1}$) as a function of SF ($f$) at each eccentricity ($r$) and polar angle ($\theta$) using three or four parameters (*Table 1*).

### Visual field models

We implemented six models at the group level, specifying how CSFs change across eccentricity and polar angle. For each model, we iteratively fixed the CSF's parameters to permit or restrict the impact of eccentricity, HVA, and/or VMA on contrast sensitivity (*Figure 3*). For example, the most permissive model ('Ecc + HVA + VMA,' *Table 2*) allowed CSFs to vary freely across eccentricity and polar angle, which yielded 24 parameters for CSF models with four parameters (e.g., YQM model, *Table 1*; 4 parameters × 2 eccentricities × 3 polar angles = 24 parameters, *Table 2*). In contrast, the most restrictive model ('-Ecc – HVA – VMA,' *Table 2*) enforced a single CSF at all visual field locations, yielding only four parameters. A detailed breakdown of the model alternatives is presented in *Table 2*. We additionally assessed whether CSFs depended on the pre-cue presented to observers in the fixed-size conditions and found that they did not.

### Model fitting

Our parametric contrast sensitivity model generates the probability that an observer will correctly judge a grating's orientation as a function of contrast, SF, and visual field location (*Equation 7*). We optimized the model's parameters via maximum likelihood estimation. We considered performance at each contrast, SF, eccentricity, and polar angle as independent Bernoulli random variables and minimized the negative log-likelihood for an observer's responses using *fmincon* in the MATLAB Optimization Toolbox. This procedure maximized the power of our analyses by leveraging each data point (i.e., trial).

We performed model fitting at the group level and in two stages to (1) identify the best-fitting CSF model and (2) determine the appropriate visual field model. To identify the best CSF model, we fit each CSF model (*Table 1*) to the group-level behavioral responses across all three conditions (HM, VM, and M-scale). For these fits, the CSFs followed the most permissive visual field model (Ecc + HVA + VMA, *Table 2*). Model comparisons determined the best-fitting CSF model (see 'Model comparisons').

After identifying the best CSF model, we determined which visual field model corresponded best to the observers' contrast sensitivity across eccentricity and polar angle. To this end, we fit each visual field model (*Table 2*) to the group-level responses to fixed-size grating stimuli (HM and VM conditions) because these stimuli yield robust variations in contrast sensitivity across eccentricity and polar angle (*Abrams et al., 2012*; *Cameron et al., 2002*; *Carrasco et al., 2001*; *Himmelberg et al., 2020*; *Jigo and Carrasco, 2020*). For these fits, we used the best-fitting CSF model identified in stage 1.

## Model comparisons

We compared CSF models (*Table 1*) and visual field models (*Table 2*). The difference in BIC values between model variants indexed model performance, with lower values corresponding to better performance. We calculated BIC values as $BIC = 2klog(n)L$, where $k$ denotes the number of model parameters, $n$ denotes the number of trials, and $L$ corresponds to a model's maximized log-likelihood.

## Quantifying the extent of eccentricity effects and polar angle asymmetries

We quantified the impact of changing visual field location (e.g., 2° to 6°) as the percent change in contrast sensitivity ($\Delta c_t^{-1}$) between one location ($l_1$) and the other ($l_2$), normalized by the average contrast sensitivity among locations (*Equation 9*):

$$\Delta c_t^{-1} = 100\frac{l_1-l_2}{0.5(l_1+l_2)} \tag{9}$$

## CSF attributes

For each observer, we extracted key CSF attributes from the best-fitting CSF model: peak-CS, peak-SF, cutoff-SF, AULCSF, and SF bandwidth. Because not all CSF models in *Table 1* have parameters that map onto these attributes, we evaluated the CSF between 0.25 cpd and 24 cpd. We defined the peak-CS as the maximum contrast sensitivity of the CSF, the peak-SF as the SF at which peak-CS occurred, the cutoff-SF as the SF at which contrast sensitivity reached its minimum value of 1, and the SF bandwidth as the number of octaves spanned at the CSF's full-width-at-half-maximum.

## fMRI analysis

We were able to obtain seven observers population receptive field (pRF) (*Dumoulin and Wandell, 2008*) and anatomical data from the NYU Retinotopy Database (*Himmelberg et al., 2021*). These retinotopy data were used to calculate the amount of V1 surface area representing the HM, LVM, and UVM in each observer's V1 map. The pRF stimulus, MRI and fMRI acquisition parameters, MRI and fMRI preprocessing (*Esteban et al., 2019*), the implementation of the pRF model, and the computation of the amount of V1 surface area represented by wedge-ROIs centered on the cardinal meridians of the visual field, were identical to the methods described in our prior work (*Himmelberg et al., 2022b*; *Himmelberg et al., 2021*). In brief, we calculated the amount of V1 surface area representing the HM (mean of left and right HM), the UVM, and the LVM by defining ±15° wedge-ROIs in the visual field that were centered along the four polar angle meridians. Each wedge-ROI extended from 1 to 8° of eccentricity. The amount of V1 surface area encapsulated by these wedge-ROIs was calculated by summing the surface (mm²) of the vertices that had pRF centers within these wedge-ROIs. The output of this analysis is the amount of V1 surface area (mm²) representing the wedge-ROIs at each meridian. Any differences in V1 surface area derived from these wedge-ROIs can be considered to index differences in cortical magnification – either among meridians or among observers.

## Statistical analyses

We used repeated-measures ANOVAs followed by paired *t*-tests for post hoc comparisons. All post hoc comparisons were Bonferroni-corrected for multiple comparisons. All p-values for repeated-measures ANOVAs in which the assumption of sphericity was not met were Greenhouse–Geisser

corrected. Each ANOVA assessed how M-scaling affected the extent of eccentricity effects and polar angle asymmetries. We used separate ANOVAs to assess how M-scaling affected the perifoveal HVA and VMA. We report effect sizes in terms of generalized eta squared ($\eta_G^2$) for ANOVAs and Cohen's d for $t$-tests.

## Acknowledgements

The authors thank Antoine Barbot, Antonio Fernández, Nina Hanning, Shutian Xue, and other members of the Carrasco lab, as well as Michael Landy, for their helpful comments and discussion. This work was supported by the National Institutes of Health R01-EY027401 to MC.

## Additional information

### Competing interests

Marisa Carrasco: Reviewing editor, *eLife*. The other authors declare that no competing interests exist.

### Funding

| Funder | Grant reference number | Author |
|---|---|---|
| National Eye Institute | R01-EY027401 | Marisa Carrasco |

The funders had no role in study design, data collection and interpretation, or the decision to submit the work for publication.

### Author contributions

Michael Jigo, Conceptualization, Investigation, Visualization, Methodology, Writing - original draft, Writing – review and editing; Daniel Tavdy, Investigation, Methodology, Writing – review and editing; Marc M Himmelberg, Formal analysis, Investigation, Visualization, Methodology, Writing – review and editing; Marisa Carrasco, Conceptualization, Supervision, Funding acquisition, Investigation, Methodology, Writing - original draft, Project administration, Writing – review and editing

### Author ORCIDs

Michael Jigo  http://orcid.org/0000-0002-9742-4576
Daniel Tavdy  http://orcid.org/0009-0003-0431-1308
Marc M Himmelberg  http://orcid.org/0000-0001-9133-7984
Marisa Carrasco  http://orcid.org/0000-0002-1002-9056

### Ethics

Human subjects: All observers provided written informed consent under the University Committee's protocol on Activities Involving Human Subjects at New York University agreeing to participate in the study and the public release of their data. All experimental procedures were approved by the Ethics Committee at the NYU Department of Psychology (IRB: FY2016-466) and were in agreement with the Declaration of Helsinki.

### Decision letter and Author response

Decision letter https://doi.org/10.7554/eLife.84205.sa1
Author response https://doi.org/10.7554/eLife.84205.sa2

## Additional files

### Supplementary files
• MDAR checklist

### Data availability

Data and code pertaining to the experiment are available on the OSF repository (https://osf.io/gvkdh/; Jigo et al., 2023).

The following dataset was generated:

| Author(s) | Year | Dataset title | Dataset URL | Database and Identifier |
|---|---|---|---|---|
| Jigo M, Tavdy D, Himmelberg MM, Carrasco M | 2023 | Mscaling eliminates contrast sensitivity across not around | https://osf.io/gvkdh/ | Open Science Framework, 10.17605/OSF.IO/GVKDH |

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
