## [Editor Report]

This important study presents a thought-provoking challenge to the explanation of sensitivity around the visual field using cortical magnification factors. The evidence supporting this challenge is based on a combination of neuroimaging and psychophysics. The study will be of interest to both basic and medical vision researchers.

---

## [Decision Letter]

**Decision letter after peer review:**

Thank you for submitting your article "Cortical magnification underlies differences across but not around the visual field" for consideration by *eLife*. Your article has been reviewed by 3 peer reviewers, and the evaluation has been overseen by a Reviewing Editor and Chris Baker as the Senior Editor. The following individuals involved in the review of your submission have agreed to reveal their identity: Jiawei Zhou (Reviewer #1); Dietrich Samuel Schwarzkopf (Reviewer #3).

Essential revisions:

1) Provide subject-specific measurements of cortical magnification factors. For instance, as reviewer #3 stated and agreed by the other two reviewers, the scaling for the upper and lower vertical meridian is almost identical. This is based on the equations given in a previous study (Rovamo and Virsu, 1979) and it is unclear whether those estimates are fully appropriate. A better approach might have been to scale stimuli based on each individual's own cortical magnification factor, or at least on a group average of the sample of 10 tested here.

2) Add or extend discussions to interpret possible mechanisms. Point-to-point responses addressing all 3 reviewers' comments on the discussion part should be provided.

3) Provide more methodological details.

A) The description of the cortical magnification component of the methods, which is quite important, could be expanded on a bit more, or even placed in the body of the main text, given its importance. Incidentally, it was difficult to figure out what the references were in the Methods because they were indexed using a numbering system (formatted for perhaps a different journal).

B) Another methodological aspect of the study that was unclear was how the fitting worked. The authors do a commendably thorough job incorporating numerous candidate CSF models. However, it seems that each participant was fitted with all the models, and the best model was then used to test the various anisotropy models afterwards. What was the motivation for letting each individual have their own qualitatively distinct CSF model? That seems rather unusual. Related to this, while the peak of the CSF is nicely sampled, there was a lack of much data in the cutoff at higher spatial frequencies, which at least in the single subject data that was shown made the cutoff frequency measure seem like it would be unreliable. Did the authors find that to be an issue in fitting the data?

C) Psychophysical tests are very demanding on display devices. Considering the maximum contrast sensitivity of 200, was the monitor calibrated with high grayscale or only with 8-bit? Also, was the global uniformity of the display calibrated when measured at different locations?

D) Please add the number of trials corresponding to each SF in each CSF curve to the method.

E) In Figure 6, it is desirable to add panels of the exact values of the HVA and VMA effects for key CSF attributes at different eccentricities.

4) Address potential confounds.

A) Due to the different testing distances in VM and HM, their retinae will be in a different adaptation state, making any comparison between VM and HM tricky.

B) In addition to the key CSF attributes used in this paper, the area under the CSF curve is a common, global parameter to figure out how contrast sensitivity changes under different conditions. An analysis of the area under the CSF curve is recommended.

C) In Figure 4, the HVA extent appears to change after M-scaling, although the analysis shows that M-scaling only affects the HVA extent at high SF. In contrast, the range of VMA was almost unchanged.

D) The author suggested that the apparent reduction in the HVA extent at high SF may be due to the lower cutoff SF of the perifoveal VM. Analysis of the correlation between the change in HVA and the cutoff SF after M scaling may help to draw more comprehensive conclusions.

E) The results in Figure 4 also show that at 11.3 cpd, the measurement may be inaccurate. This might lead to an inaccurate estimate of the M scaling effect at 11.3 cpd.

*Reviewer #1 (Recommendations for the authors):*

In this article, the authors used classic psychophysical tests and a simple experimental design to answer the question of whether cortical magnification underlies polar angle asymmetries of contrast sensitivity. Contrast sensitivity is considered to be the most fundamental spatial vision and is important for both normal individuals and clinical patients in ophthalmology. The parametric contrast sensitivity model and the extraction of key CSF attributes help to compare the comparison of the effect of M scaling at different angles. This work can provide a new reference for the study of normal and abnormal space vision.

I just have the following comments that I would like the authors to address:

Methods:

1) Psychophysical tests are very demanding on display devices. Considering the maximum contrast sensitivity of 200, was the monitor calibrated with high grayscale or only with 8-bit? Also, was the global uniformity of the display calibrated when measured at different locations?

2) Please add the number of trials corresponding to each SF in each CSF curve to the method.

Results:

1) In Figure 2, several CRFs for SF are given but were the CRFs at the cutoff-SF well-fitted? Please provide the results of CRF and the corresponding goodness of fit.

2) The analysis of the area under the CSF curve is recommended.

3) The author suggested that the apparent reduction in the HVA extent at high SF may be due to the lower cutoff-SF of the perifoveal VM. Analysis of the correlation between the change in HVA and the cutoff SF after M scaling may help to draw more comprehensive conclusions.

4) In Figure 6, it is desirable to add panels of the exact values of the HVA and VMA effects for key CSF attributes at different eccentricities, as shown in Figures 4B, D, and F shown, to make the results more intuitive.

Discussion:

1) Due to the different testing distances in VM and HM, their retinae will be in a different adaptation state, making any comparison between VM and HM tricky. Please add a discussion on this issue.

2) In Figure 4, the HVA extent appears to change after M-scaling, although the analysis shows that M-scaling only affects the HVA extent at high SF. In contrast, the range of VMA was almost unchanged. The authors could have discussed more about how the HVA and VMA effects behave differently after M-scaling.

3) The results in Figure 4 also show that at 11.3 cpd, the measurement may be inaccurate. This might lead to an inaccurate estimate of the M scaling effect at 11.3 cpd. The authors should discuss this issue more.

*Reviewer #3 (Recommendations for the authors):*

As explained in the public review, it seems important to consider whether the M-scaling used at the different meridians was sufficient to correct cortical magnification differences. The scaling for the upper and lower vertical meridian is almost identical. This is based on the equations given in a previous study (Rovamo and Virsu, 1979) and it is unclear whether those estimates are fully appropriate. A better approach might have been to scale stimuli based on each individual's own cortical magnification factor, or at least on a group average of the sample of 10 tested here. I realise that this entails further experiments, but it at least seems that computing the cortical magnification should be relatively straightforward: as far as I understood the experiments here are based on a sample used in the authors' previous work (Himmelberg et al., 2022), although there is some confusion about references (see below). In lieu of carrying out new measurements, it could also suffice to compare individual cortical magnification factors to the performance to quantify the contribution to the psychophysical performance.

In terms of discussing possible mechanisms, and even putting the work into a broader context, more of the background literature deserves to be discussed (I openly admit that some of these studies are our own – I wouldn't suggest including them if I didn't feel they were relevant). For starters, we have previously shown that visual object size perception displays idiosyncratic biases and variations in discrimination ability (possibly related to acuity) across the visual field (Moutsiana et al., 2016). Part of these are group average differences between visual field meridians (albeit not cardinal ones) but there are also substantial individual differences between observers. We found that perceptual performance in locations was correlated with pRF size in V1, which in turn is also linked with cortical magnification (V1 surface area) – seen e.g. in Figure 7 of that study and previously reported by other work (Harvey and Dumoulin, 2011; Song et al., 2015). Moreover, neuroimaging work by another lab has shown that pRF size and cortical magnification in the human visual cortex vary between visual field meridians (Silva et al., 2017). Based on this finding, my lab tested the visual object size biases between meridians and found stronger perceptual biases (and acuity differences) between horizontal and vertical meridians (Schwarzkopf, 2019). Similar idiosyncrasies in perceptual functions between meridians and visual field locations have also been reported in several other studies (Afraz et al., 2010; Finlayson et al., 2020; Greenwood et al., 2017; Kosovicheva and Whitney, 2017; Wang et al., 2020).

References

Afraz, A., Pashkam, M.V., Cavanagh, P., 2010. Spatial heterogeneity in the perception of face and form attributes. Curr. Biol 20, 2112-2116. https://doi.org/10.1016/j.cub.2010.11.017

Duncan, R.O., Boynton, G.M., 2003. Cortical magnification within human primary visual cortex correlates with acuity thresholds. Neuron 38, 659-671.

Finlayson, N.J., Neacsu, V., Schwarzkopf, D.S., 2020. Spatial Heterogeneity in Bistable Figure-Ground Perception: i-Perception. https://doi.org/10.1177/2041669520961120

Greenwood, J.A., Szinte, M., Sayim, B., Cavanagh, P., 2017. Variations in crowding, saccadic precision, and spatial localization reveal the shared topology of spatial vision. PNAS 114, E3573-E3582. https://doi.org/10.1073/pnas.1615504114

Harvey, B.M., Dumoulin, S.O., 2011. The Relationship between Cortical Magnification Factor and Population Receptive Field Size in Human Visual Cortex: Constancies in Cortical Architecture. J. Neurosci. 31, 13604-13612. https://doi.org/10.1523/JNEUROSCI.2572-11.2011

Himmelberg, M.M., Winawer, J., Carrasco, M., 2022. Linking individual differences in human primary visual cortex to contrast sensitivity around the visual field. Nat Commun 13, 3309. https://doi.org/10.1038/s41467-022-31041-9

Kosovicheva, A., Whitney, D., 2017. Stable individual signatures in object localization. Curr. Biol. 27, R700-R701. https://doi.org/10.1016/j.cub.2017.06.001

Moutsiana, C., de Haas, B., Papageorgiou, A., van Dijk, J.A., Balraj, A., Greenwood, J.A., Schwarzkopf, D.S., 2016. Cortical idiosyncrasies predict the perception of object size. Nat Commun 7, 12110. https://doi.org/10.1038/ncomms12110

Rovamo, J., Virsu, V., 1979. An estimation and application of the human cortical magnification factor. Exp Brain Res 37, 495-510. https://doi.org/10.1007/BF00236819

Schwarzkopf, D.S., 2019. Size Perception Biases Are Temporally Stable and Vary Consistently Between Visual Field Meridians. i-Perception 10, 2041669519878722. https://doi.org/10.1177/2041669519878722

Schwarzkopf, D.S., Rees, G., 2013. Subjective size perception depends on central visual cortical magnification in human v1. PLoS ONE 8, e60550. https://doi.org/10.1371/journal.pone.0060550

Schwarzkopf, D.S., Song, C., Rees, G., 2011. The surface area of human V1 predicts the subjective experience of object size. Nat. Neurosci 14, 28-30. https://doi.org/10.1038/nn.2706

Silva, M.F., Brascamp, J.W., Ferreira, S., Castelo-Branco, M., Dumoulin, S.O.,

Harvey, B.M., 2017. Radial asymmetries in population receptive field size and cortical magnification factor in early visual cortex. Neuroimage 167, 41-52. https://doi.org/10.1016/j.neuroimage.2017.11.021

Song, C., Schwarzkopf, D.S., Kanai, R., Rees, G., 2015. Neural Population Tuning Links Visual Cortical Anatomy to Human Visual Perception. Neuron 85, 641-56. https://doi.org/10.1016/j.neuron.2014.12.041

Wang, Z., Murai, Y., Whitney, D., 2020. Idiosyncratic perception: a link between acuity, perceived position and apparent size. Proc Biol Sci 287, 20200825. https://doi.org/10.1098/rspb.2020.0825

---

## [Author Response]

Essential revisions:1) Provide subject-specific measurements of cortical magnification factors. For instance, as reviewer #3 stated and agreed by the other two reviewers, the scaling for the upper and lower vertical meridian is almost identical. This is based on the equations given in a previous study (Rovamo and Virsu, 1979) and it is unclear whether those estimates are fully appropriate. A better approach might have been to scale stimuli based on each individual's own cortical magnification factor, or at least on a group average of the sample of 10 tested here.

– We note that the equations by Rovamo and Virsu are commonly used to cortically magnify stimulus size. This paper has many citations, and the conclusions of many studies are based on those calculations [lines 115-128].

– In response to Rev’s 3 comment, “In lieu of carrying out new measurements, it could also suffice to compare individual cortical magnification factors to the performance to quantify the contribution to the psychophysical performance”, we found a significant correlation between the surface area and contrast sensitivity measures at the horizontal, upper-vertical and lower-vertical meridians. However, we found no significant correlation between the cortical surface with the difference in contrast sensitivity for fixed-size and magnified stimuli at 6 deg at each meridian. These findings suggest that surface area plays a role but that individual magnification is unlikely to equalize contrast sensitivity [lines 366-380; Figure 7; lines 511-529].

2) Add or extend discussions to interpret possible mechanisms. Point-to-point responses addressing all 3 reviewers' comments on the discussion part should be provided.

– We have expanded the discussion of qualitative hypothesis of differences in polar angle [lines 86-92; lines 476-481].

– We have expanded the discussion of possible mechanisms [lines 496-529].

– We have explained why having assessed the VM and HM and different distances does not significantly influence our measures [lines 483-491].

– We have expanded the discussion of how the HVA and VMA effects behave differently after M-scaling [lines 435-450].

– We have clarified that the fits are reliable and made explicit that the highest SF data point is at chance in both conditions [FIGURE 4 caption].

3) Provide more methodological details. A) The description of the cortical magnification component of the methods, which is quite important, could be expanded on a bit more, or even placed in the body of the main text, given its importance. Incidentally, it was difficult to figure out what the references were in the Methods because they were indexed using a numbering system (formatted for perhaps a different journal).

We now detail M-scaling in the Introduction [lines 115-135]**,** and we have fixed the references in the Methods section.

B) Another methodological aspect of the study that was unclear was how the fitting worked. The authors do a commendably thorough job incorporating numerous candidate CSF models. However, it seems that each participant was fitted with all the models, and the best model was then used to test the various anisotropy models afterwards. What was the motivation for letting each individual have their own qualitatively distinct CSF model? That seems rather unusual. Related to this, while the peak of the CSF is nicely sampled, there was a lack of much data in the cutoff at higher spatial frequencies, which at least in the single subject data that was shown made the cutoff frequency measure seem like it would be unreliable. Did the authors find that to be an issue in fitting the data?

– We have further clarified that we fit all 9 models to the grouped data [lines 177-178] and in Methods [lines 693, 716, 725], and that the fit in Figure 3 corresponds to the grouped data [Figure 3 caption]. As reported in Figure 4A,C,E, the group data fits were very high (≥.98). Please note that the cutoff spatial frequency is reliable. The data point (11.3 cpd) in the differences which does not follow the same function (Figure 4D,F) reflects the fact that for both magnified and not-magnified stimuli, performance was at chance, consistent with the fact that high SF are harder to discriminate at peripheral locations [Figure 4 caption].

C) Psychophysical tests are very demanding on display devices. Considering the maximum contrast sensitivity of 200, was the monitor calibrated with high grayscale or only with 8-bit? Also, was the global uniformity of the display calibrated when measured at different locations?

– The monitor was calibrated with 8-bit at the center of the display [lines 607].

– Regarding global uniformity, although we only calibrated at the center of the display, please note that the asymmetries are not due to the particular monitor we used. We have obtained these asymmetries in contrast sensitivity in numerous studies using multiple monitors over 20 years (e.g., Carrasco, Talgar and Cameron, 2001; Cameron, Tai and Carrasco, 2002; Fuller, Park and Carrasco, 2009; Abrams, Nizam and Carrasco, 2012; Corbett and Carrasco, 2012; Hanning et al., 2022a; Himmelberg et al., 2020) and other groups have reported these visual asymmetries as well (Baldwin et al., 2012; Pointer and Hess, 1989; Rosén et al., 2014). Also important, as we had mentioned in the Introduction [lines 55-59], the HVA and VMA asymmetries shift in-line with egocentric referents, corresponding to the retinal location of the stimulus, not with the allocentric location (Corbett and Carrasco, 2011).

D) Please add the number of trials corresponding to each SF in each CSF curve to the method.

– We have done so [lines 637-644]

E) In Figure 6, it is desirable to add panels of the exact values of the HVA and VMA effects for key CSF attributes at different eccentricities.

– We have added these panels [FIGURE 6] and the corresponding analysis in the text [lines 321-343].

4) Address potential confounds. A) Due to the different testing distances in VM and HM, their retinae will be in a different adaptation state, making any comparison between VM and HM tricky.

– Note that the mean luminance of the display (from retina to monitor) was 23 cd/m^2^ at 57cm and 19 cd/m^2^ at 115 cm. The pupil size difference for these two conditions is relatively small (< 0.5 mm) and should not significantly affect contrast sensitivity (Rahimi-Nasrabadi et al., 2021) [lines 483-491]. Moreover, the differences we get here are consistent with the asymmetries we (e.g., Carrasco, Talgar and Cameron, 2001; Cameron, Tai and Carrasco, 2002; Fuller, Park and Carrasco, 2009; Abrams, Nizam and Carrasco, 2012; Corbett and Carrasco, 2012; Himmelberg, Winawer and Carrasco, 2020) and many others (e.g., Baldwin et al., 2012; Pointer and Hess, 1989; Regan and Beverley, 1983; Rijsdijk et al., 1980; Robson and Graham, 1981; Rosén et al., 2014; Silva et al., 2008) have observed for contrast sensitivity when the vertical and horizontal meridian are tested simultaneously at the same distance.

B) In addition to the key CSF attributes used in this paper, the area under the CSF curve is a common, global parameter to figure out how contrast sensitivity changes under different conditions. An analysis of the area under the CSF curve is recommended.

– We have added the area under the CSF (AULCSF) [lines 304-318, Figure 5 E-F; lines 338-343, Figure 6 E-F]. Differences for non-magnified and magnified stimuli are not eliminated.

C) In Figure 4, the HVA extent appears to change after M-scaling, although the analysis shows that M-scaling only affects the HVA extent at high SF. In contrast, the range of VMA was almost unchanged.

– We had commented on this pattern and have further clarified [lines 436-451].

D) The author suggested that the apparent reduction in the HVA extent at high SF may be due to the lower cutoff SF of the perifoveal VM. Analysis of the correlation between the change in HVA and the cutoff SF after M scaling may help to draw more comprehensive conclusions.

– As per the previous point, we have rephrased our explanation [lines 453-460].

E) The results in Figure 4 also show that at 11.3 cpd, the measurement may be inaccurate. This might lead to an inaccurate estimate of the M scaling effect at 11.3 cpd.

– We have explained why this data point is at chance [FIGURE 4 caption].

Reviewer #1 (Recommendations for the authors):In this article, the authors used classic psychophysical tests and a simple experimental design to answer the question of whether cortical magnification underlies polar angle asymmetries of contrast sensitivity. Contrast sensitivity is considered to be the most fundamental spatial vision and is important for both normal individuals and clinical patients in ophthalmology. The parametric contrast sensitivity model and the extraction of key CSF attributes help to compare the comparison of the effect of M scaling at different angles. This work can provide a new reference for the study of normal and abnormal space vision.I just have the following comments that I would like the authors to address:

Please note that we have addressed all the points below, as they were included in the Essential Revisions

Methods:1) Psychophysical tests are very demanding on display devices. Considering the maximum contrast sensitivity of 200, was the monitor calibrated with high grayscale or only with 8-bit? Also, was the global uniformity of the display calibrated when measured at different locations?

– see 3C, essential revisions

2) Please add the number of trials corresponding to each SF in each CSF curve to the method.

– see 3D, essential revisions

Results:1) In Figure 2, several CRFs for SF are given but were the CRFs at the cutoff-SF well-fitted? Please provide the results of CRF and the corresponding goodness of fit.

– see 2, essential revisions

2) The analysis of the area under the CSF curve is recommended.

– see 4B, essential revisions

3) The author suggested that the apparent reduction in the HVA extent at high SF may be due to the lower cutoff-SF of the perifoveal VM. Analysis of the correlation between the change in HVA and the cutoff SF after M scaling may help to draw more comprehensive conclusions.

– see 4D, essential revisions

4) In Figure 6, it is desirable to add panels of the exact values of the HVA and VMA effects for key CSF attributes at different eccentricities, as shown in Figures 4B, D, and F shown, to make the results more intuitive.

– see 3E, essential revisions

Discussion:1) Due to the different testing distances in VM and HM, their retinae will be in a different adaptation state, making any comparison between VM and HM tricky. Please add a discussion on this issue.

– see 4A, essential revisions

2) In Figure 4, the HVA extent appears to change after M-scaling, although the analysis shows that M-scaling only affects the HVA extent at high SF. In contrast, the range of VMA was almost unchanged. The authors could have discussed more about how the HVA and VMA effects behave differently after M-scaling.

– see 4C, essential revisions

3) The results in Figure 4 also show that at 11.3 cpd, the measurement may be inaccurate. This might lead to an inaccurate estimate of the M scaling effect at 11.3 cpd. The authors should discuss this issue more.

– see 4E, essential revisions

Reviewer #3 (Recommendations for the authors):As explained in the public review, it seems important to consider whether the M-scaling used at the different meridians was sufficient to correct cortical magnification differences. The scaling for the upper and lower vertical meridian is almost identical. This is based on the equations given in a previous study (Rovamo and Virsu, 1979) and it is unclear whether those estimates are fully appropriate. A better approach might have been to scale stimuli based on each individual's own cortical magnification factor, or at least on a group average of the sample of 10 tested here. I realise that this entails further experiments, but it at least seems that computing the cortical magnification should be relatively straightforward: as far as I understood the experiments here are based on a sample used in the authors' previous work (Himmelberg et al., 2022), although there is some confusion about references (see below). In lieu of carrying out new measurements, it could also suffice to compare individual cortical magnification factors to the performance to quantify the contribution to the psychophysical performance.

– see 1, essential revisions

In terms of discussing possible mechanisms, and even putting the work into a broader context, more of the background literature deserves to be discussed (I openly admit that some of these studies are our own – I wouldn't suggest including them if I didn't feel they were relevant). For starters, we have previously shown that visual object size perception displays idiosyncratic biases and variations in discrimination ability (possibly related to acuity) across the visual field (Moutsiana et al., 2016). Part of these are group average differences between visual field meridians (albeit not cardinal ones) but there are also substantial individual differences between observers. We found that perceptual performance in locations was correlated with pRF size in V1, which in turn is also linked with cortical magnification (V1 surface area) – seen e.g. in Figure 7 of that study and previously reported by other work (Harvey and Dumoulin, 2011; Song et al., 2015). Moreover, neuroimaging work by another lab has shown that pRF size and cortical magnification in the human visual cortex vary between visual field meridians (Silva et al., 2017). Based on this finding, my lab tested the visual object size biases between meridians and found stronger perceptual biases (and acuity differences) between horizontal and vertical meridians (Schwarzkopf, 2019). Similar idiosyncrasies in perceptual functions between meridians and visual field locations have also been reported in several other studies (Afraz et al., 2010; Finlayson et al., 2020; Greenwood et al., 2017; Kosovicheva and Whitney, 2017; Wang et al., 2020).

– see 2, essential revisions

– Throughout Introduction and discussion, we have integrated relevant studies of polar angle, and added discussion of possible mechanisms [lines 497-530].